# LANGUAGE MODELS ARE GREEDY REASONERS: A SYSTEMATIC FORMAL ANALYSIS OF CHAIN-OF-THOUGHT

**Abulhair Saparov & He He**
Center for Data Science, New York University, New York, NY 10011, USA
{as17582,hhe}@nyu.edu

## ABSTRACT

Large language models (LLMs) have shown remarkable reasoning capabilities given chain-of-thought prompts (examples with intermediate reasoning steps). Existing benchmarks measure reasoning ability indirectly, by evaluating accuracy on downstream tasks such as mathematical reasoning. However, it is unclear *how* these models obtain the answers and whether they rely on simple heuristics rather than the generated chain-of-thought. To enable systematic exploration of the reasoning ability of LLMs, we present a new synthetic question-answering dataset called PRONTOQA, where each example is generated from a synthetic world model represented in first-order logic. This allows us to parse the generated chain-of-thought into symbolic proofs for formal analysis. Our analysis on INSTRUCTGPT and GPT-3 shows that LLMs are quite capable of making correct individual deduction steps, and so are generally capable of reasoning, even in fictional contexts. However, they have difficulty with *proof planning*: When multiple valid deduction steps are available, they are not able to systematically explore the different options.

## 1 INTRODUCTION

The ability to reason—drawing new conclusions from provided facts—is a hallmark of human intelligence. Recently, *chain-of-thought* (CoT) prompting has enabled large language models (LLMs) to perform logical reasoning tasks with impressive accuracy (Wei et al., 2022; Chowdhery et al., 2022; Lewkowycz et al., 2022). In CoT prompting, each example consists of a *question* (e.g., "$\frac{6}{3} - 1$?"), a short description of the reasoning required to answer the question called the *"chain-of-thought"* (e.g., "$\frac{6}{3}$ is 2. $2 - 1$ is 1."), and a *label* (e.g., "1"). When prompted with a few CoT examples, the elicited reasoning allows LLMs to predict the label with much higher accuracy than standard question-answer prompting. However, it is unclear to what extent these models can reason due to several confounding factors. First, existing studies primarily rely on question-answering (QA) tasks from real-world settings such as math word problems (Cobbe et al., 2021; Han et al., 2022; Weston et al., 2016). It is likely that LLMs have already acquired the knowledge through pretraining and simply retrieve the answer rather than reason over it. Second, the reasoning task may contain spurious correlations that allow the model to obtain the correct answer through shortcuts (Zhang et al., 2022b). In this work, we systematically investigate the reasoning capability of LLMs by directly evaluating their predicted chains-of-thought (the interpretable proof steps), rather than the predicted label.

To enable easy analysis of the CoT, we construct a new synthetic QA dataset called PRONTOQA, for **Pr**oof and **Onto**logy-Generated **Q**uestion-**A**nswering. Inspired by the PROOFWRITER dataset (Tafjord et al., 2021), each example in PRONTOQA is generated from an ontology and has a unique proof (see figure 1 for an example). We convert the proofs into syntactically simple sentences using a grammar such that the inverse process is relatively easy: From the predicted CoT, we semantically parse each sentence into a formal language and reconstruct the underlying proof steps. We then directly analyze the model's reasoning by inspecting each step in the reconstructed proof and comparing them against the gold proof.[1] We emphasize here that while the dataset is an important contribution of this paper, the main contribution is the analysis that is facilitated by the dataset.

---

[1]All analysis code, data, data generation scripts, and model outputs are available at github.com/asaparov/prontoqa.

**Q:** *Each cat is a carnivore. Every carnivore is not herbivorous. Carnivores are mammals. All mammals are warm-blooded. Mammals are vertebrates. Every vertebrate is an animal. Animals are multicellular.* —— context
*Fae is a cat. True or false: Fae is not herbivorous.* —— query
**A:** *Fae is a cat. Cats are carnivores. Fae is a carnivore. Every carnivore is not herbivorous.* —— chain-of-thought
*Fae is not herbivorous.* *True* —— label

FIGURE 1: A question-answering example from PRONTOQA, with each component highlighted and labeled.

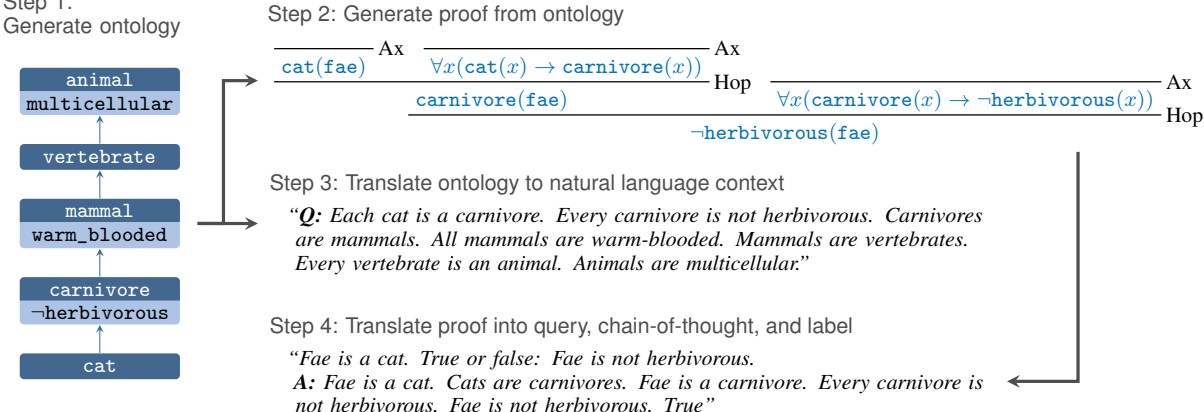

FIGURE 2: Schematic of the generative process for each example in PRONTOQA. **Step 1:** We generate an ontology from a prior distribution, shown here as a tree. Each node denotes a concept (e.g., `mammal`), each with an optional property (e.g., `warm_blooded`), and each blue edge denotes a "subtype of" relation. **Step 2:** Generate proof from the ontology. Each horizontal black line indicates a proof step, with its premises written above the line and the conclusion written below. **Step 3:** Convert the ontology into a natural language context. **Step 4:** Convert the proof into a natural language query, chain-of-thought, and answer label. There is a one-to-one correspondence between the conclusion of each proof step and the sentences in the chain-of-thought.

We systematically evaluate INSTRUCTGPT[2] (Ouyang et al., 2022) and the original GPT-3 (Brown et al., 2020) on PRONTOQA by controlling a number of variables that characterize the complexity of the reasoning task, such as the ontology type and the number of proof steps required. Our analysis shows that these models are quite good at producing valid *individual* proof steps, even on fictional and counterfactual ontologies. However, LLMs have difficulty with *proof planning*: when the models encounter a point in the proof where multiple valid proof steps are available, they sometimes select the wrong step, and this often leads to an incomplete proof and subsequently an incorrect answer. Interestingly, the models are much less likely to be misled with a true ontology, suggesting that the world knowledge acquired during pretraining plays an important role in LLM reasoning. We also find that our results generalize to more sophisticated/informative prompts, such as *self-consistency* prompting (Wang et al., 2022), and prompts with example traces of depth-first proof search instead of CoT.

## 2 RELATED WORK

Our proposed dataset is most closely related to PROOFWRITER (Tafjord et al., 2021) and FOLIO (Han et al., 2022) which are QA datasets designed to test reasoning ability. PROOFWRITER provides multi-hop proofs for each example. However, there are a number of key properties that led us to develop our own dataset (see table 1 for a summary). FOLIO does not provide easily-parseable proofs/CoTs in their examples, and evaluation is done by inspecting the predicted labels, which may not necessarily be a good measure of reasoning ability. In our analysis, we focus on more specific variables that may affect the reasoning of the model, such as: (1) Is the model's reasoning dependent on whether the example is consistent with pretraining ("true"), inconsistent ("false"), or neither ("fictional")? (2) Is the model's reasoning sensitive to whether the predicates in the examples are unary or binary? (3) Is the model's reasoning dependent on the rules of deduction in the examples? These variables are not controllable in existing datasets. Further, in some datasets, the code to generate examples is not available.

---

[2]INSTRUCTGPT is the model resulting from fine-tuning GPT-3 via reinforcement learning from human feedback. Throughout the paper, "INSTRUCTGPT" refers to the model named `text-davinci-002`. But note that in our experiments, we also evaluate `text-ada-001`, `text-babbage-001`, `text-curie-001`, `davinci`, and `text-davinci-001`.

| Dataset | Provides easily semantically-parseable proofs | Controls for true vs false vs fictional contexts | Controls for unary vs binary predicates | Controls for specific rules of deduction | Tests reasoning beyond the domain of math word problems | Generation code available |
|---|---|---|---|---|---|---|
| GSM8K Cobbe et al. (2021) | ✗ | ✗ | ✗ | ✗ | ✗ | human-annotated |
| ProofWriter Tafjord et al. (2021) | ✓ | ✗ | ✗ | ∼ | ✓ | ✗ |
| FOLIO Han et al. (2022) | ✗ | ✗ | ✗ | ✗ | ✓ | human-annotated |
| SimpleLogic Zhang et al. (2022b) | ✗ | ✗ | ✓ | ✓ | ✓ | ✓ |
| PRONTOQA (proposed dataset) | ✓ | ✓ | ✓ | ✓ | ✓ | ✓ |

TABLE 1: Comparison of existing datasets for the formal analysis of reasoning ability.

There are efforts to tweak or extend CoT prompting to elicit more sophisticated reasoning behavior (Creswell et al., 2022; Wang et al., 2022; Creswell & Shanahan, 2022; Anil et al., 2022; Dohan et al., 2022), and they have shown that these prompting extensions to CoT can improve the elicited reasoning behavior of LLMs, even with smaller models. Rather than presenting a new prompting approach, the goal of this work is to measure the reasoning ability elicited by CoT. There are other datasets that have been designed or used to measure the reasoning capabilities of transformer-based models and LLMs (Han et al., 2022; Weston et al., 2016). They show that LLMs are able to answer questions that require reasoning in the few-shot setting with reasonable accuracy. Similar to our approach, Betz (2020) converts logical forms into fairly simple natural language using templates. However, the examples in these datasets are consistent with the real-world, and so they may confound measuring reasoning ability with retrieval ability. Valmeekam et al. (2022) found that LLMs had difficulty with a fairly simple planning task, but it is not clear whether this was due to an inability to reason or other abilities instrumental in planning, such as world modeling, keeping track of state changes, and reasoning about events that occur sequentially in time. This is despite their controlling for other variables involved in planning, such as plan generation, robustness to goal formulation, among others. They experimented with examples in a "Blocksworld" environment, a significant portion of which the model can acquire from pretraining. Our work aims to address this gap. As in our approach, Dasgupta et al. (2022) specifically looked at whether LLMs can reason in fictional or counterfactual settings and found that reasoning ability is indeed negatively affected in these settings. However they did not analyze individual steps of reasoning to better understand the cause of the errors. Since we are able to formally evaluate the LLM's predicted CoT step-by-step, we are able to perform a more fine-grained analysis of their reasoning ability. Zhang et al. (2022b) showed that BERT is not able to learn to reason robustly, but they did not use CoT prompting and it is not obvious if their results generalize to LLMs, which we evaluate.

There are two broad research approaches for reasoning in NLP: (1) reasoning over a formal symbolic language, possibly with neuro-symbolic methods and/or semantic parsing (Saparov & Mitchell, 2022; Zhang et al., 2022a; Kapanipathi et al., 2021; Dong et al., 2019; Rocktäschel & Riedel, 2017), or (2) reasoning directly over natural language (Chen et al., 2021; Bostrom et al., 2022; 2021; Welleck et al., 2021; Bhagavatula et al., 2020; Angeli & Manning, 2014; MacCartney & Manning, 2009). While PRONTOQA is generated from symbolic ontologies, the examples themselves are in natural language, and so provides value to both research directions.

Recent work has examined in-context learning and found that performance on certain tasks is sensitive to the prompt (Razeghi et al., 2022; Lu et al., 2022). However, they focused on sentiment classification and simple arithmetic tasks, and it is not clear if their results generalize to reasoning. The LLM could feasibly use retrieval, rather than reasoning, to perform those tasks. Our experiments on the fictional ontology show that the model is able to reason even when there is nothing to retrieve from.

## 3 PRONTOQA: A SYNTHETIC DATASET FOR LOGICAL REASONING

We create a new dataset, called PRONTOQA for **Pr**oof and **Onto**logy-Generated **Q**uestion-**A**nswering, where each question is generated from a symbolic ontology and proof to facilitate formal analysis of the predicted CoT. To focus the scope of our exploration, and to limit the complexity of the generated questions to those within reach of current LLMs, we only consider questions that are answerable using repeated applications of the modus ponens deduction rule. More formally, *modus ponens* is a simple deduction rule where given the premises $\forall x(f(x) \rightarrow g(x))$ and $f(a)$, we conclude $g(a)$ (e.g., given "All cats are carnivores" and "Fae is a cat," we conclude "Fae is a carnivore;" see figure 6

in the appendix).[3] This rule can be easily chained together to construct proofs with controllable size.

We generate CoT examples consisting of: the context, query, CoT, and label, where the *context* is a short paragraph containing information relevant to answer the *query* (see figure 1 for an example). Each example is translated from a proof and ontology such that the inverse process is simple: the sentences in an example can be easily and uniquely parsed into symbolic logical forms amenable to formal analysis. More specifically, as shown in figure 2, we: (1) first generate an ontology from a set of concepts, (2) generate a proof by traversing the ontology, (3) translate the ontology into the natural language context, and (4) translate the proof into the query, CoT, and label by mapping logical forms to natural language sentences. We describe each step in further detail below.

**Ontology generation.**  The first step is to generate a small hierarchical *ontology*. The ontology is a set of concepts (e.g., `mammal`, `cat`, `carnivore`, etc) and subtype relations between them (e.g., $\forall x(\texttt{cat}(x) \rightarrow \texttt{carnivore}(x))$). The ontology also describes properties of concepts (e.g., $\forall x(\texttt{mammal}(x) \rightarrow \neg\texttt{cold\_blooded}(x))$). To generate questions that are not overly complex, we restrict the ontologies to be *linear* (i.e., in the tree, every node has exactly 0 or 1 child nodes). Since ontologies are randomly generated, they vary in size from as few as 3 concepts to as many as 10.

**Proof generation.**  We generate proofs from the ontology by choosing a starting node uniformly at random, and generating the initial axiom indicating that an entity has a specific type (e.g., `cat(fae)`). Then, we walk up the tree, with each step corresponding to an application of a deduction rule (i.e., a *proof step*). Each proof step consists of zero or more *premises* and one *conclusion*. We stop when we reach a node (e.g., `carnivore(fae)`), or a node property (e.g., `¬herbivorous(fae)`), such that the number of generated proof steps matches the target number of steps.

**Translation to natural language example.**  Given a generated ontology and proof, we now translate it into a natural language CoT example consisting of the question (context and query), CoT, and label. We describe how each component is generated below:

We use a simple grammar to convert the formal statements of the ontology into the natural language utterances that make up the context. Every edge in the ontology is converted into sentences such as "All cats are carnivores" or "Every cat is a carnivore." Properties of nodes are also converted into sentences of the form "All mammals are not cold-blooded," etc.

The query is generated by using the same grammar to convert the initial axiom in the proof into a natural language sentence (e.g., "Fae is a cat"). We then determine with probability 0.5 whether to ask if the conclusion of the proof is true or if its negation is false, and convert it into a natural language "true or false" query (e.g., "True or false: Fae is not herbivorous.") and label (e.g., "True").

We convert the ordered sequence of proof steps into the CoT by translating the conclusion of each proof step into a CoT sentence.

**Avoiding shortcuts.**  In section A.2 in the appendix, we describe how we add *distractor sentences* in order to remove shortcuts that would allow the model to "guess" the answer without reasoning.

A unique feature of PRONTOQA is that it is easily programmable, with a handful of tunable knobs which we use to generate examples with varying degrees of complexity and study different aspects of reasoning in LLMs. These variables are described in greater detail in section 5.1.

## 4  FORMAL ANALYSIS OF PREDICTED PROOFS

Instead of measuring the accuracy of the predicted answers (i.e., "true" or "false"), we would like to directly evaluate the predicted CoT to check if the model derives the right answer for the right reason. We endeavor to analyze whether the model is able to apply deduction rules correctly at each proof step (i.e., local correctness), but also whether the model can plan ahead and work toward proving the answer for the query (i.e., global correctness). To measure the local correctness of a given proof step, we compute whether the step follows from one or more applications of deduction rules, and whether it requires additional rules beyond those of the gold proofs. To measure the global correctness, we wish to identify proof steps that deviate from the gold proof.

To achieve this, we parse each sentence of the predicted CoT into logical form via recursive-descent parsing using the simple grammar from the generative process. We then compute whether that logical form is provable from previous logical forms via one or more applications of deduction rules. This logical form corresponds to the conclusion of a proof step. We then evaluate the correctness of each proof step by categorizing it according to three dimensions:

---

[3]In natural deduction, this rule is actually a composition of two steps: given $\forall x(f(x) \rightarrow g(x))$, use universal elimination to conclude $f(a) \rightarrow g(a)$, and given $f(a)$, use implication elimination to conclude $g(a)$.

| Step type | Example (the conclusion of each step is highlighted green) |
|---|---|
| Strictly-valid atomic correct step, or *canonical step* | *"Fae is a cat. Cats are carnivores. Fae is a carnivore. Every carnivore is not herbivorous. Fae is not herbivorous. True"* 
 **(this is the gold CoT for this example)** |
| Strictly-valid atomic misleading step | *"Fae is a cat. Cats are carnivores. Fae is a carnivore. Every carnivore is a mammal. Fae is a mammal..."* |
| Strictly-valid non-atomic correct step | *"Fae is a cat. Fae is a carnivore. Every carnivore is not herbivorous. Fae is not herbivorous. True"* |
| Strictly-valid non-atomic misleading step | *"Fae is a cat. Cats are carnivores. Fae is a carnivore. Fae is a mammal. Every mammal is a vertebrate..."* |
| Broadly-valid correct step | *"Fae is a cat. Every cat is not herbivorous. Fae is not herbivorous..."* |
| Broadly-valid misleading step | *"Fae is a cat. Every cat is a mammal. Fae is a mammal..."* |
| Invalid step | *"Fae is a cat. Cats are carnivores. Fae is a carnivore. Every carnivore is a cat. Fae is a cat..."* |

TABLE 2: The types of proof steps (and examples thereof) into which we categorize each step in the predicted chain-of-thought from LLMs. Compare the given chain-of-thought examples with the gold example provided in the first row.

1. *Validity:* Is the current proof step provable from previous steps? If it is provable using only the deduction rules that appear in the gold proofs, we say the step is *strictly-valid*. If it is provable with a more powerful proof calculus, like natural deduction, we say the step is *broadly-valid*. Otherwise, we say the step is *invalid*.
   For example, given the premises, "Cats are carnivores" and "Carnivores are mammals," the step with conclusion "Cats are mammals" is broadly-valid since an additional deduction rule is required to prove it: given $\forall x(f(x) \to g(x))$ and $\forall x(g(x) \to h(x))$, conclude $\forall x(f(x) \to h(x))$. Notice that this is distinct from a strictly-valid non-atomic step since this conclusion is not provable via repeated applications of modus ponens.We note that this the only additional rule that we check, as we did not encounter any instances of other broadly-valid rules.
2. *Atomicity:* Is the current proof step provable from previous steps with *exactly one* application of a deduction rule? If so, we say the proof step is *atomic*. Otherwise, it is *non-atomic*. Note that since all broadly-valid steps are non-atomic, this distinction is only useful for strictly-valid steps.
   For example, given the premises, "Fae is a cat," "Cats are carnivores," and "Carnivores are mammals," the step with conclusion "Fae is a mammal" is non-atomic since the step "Fae is a carnivore" was skipped.
3. *Utility:* If the current proof step's premises are part of the gold proof, but its conclusion is not, then we say the proof step is *misleading*. Otherwise, it is *correct*.
   For example, given the premises "Fae is a carnivore," "All carnivores are not herbivorous," and "Carnivores are mammals," and the goal is to prove "Fae is not herbivorous," the step "Fae is a mammal" is misleading since although the step is strictly-valid, it does not help to prove the goal.

The types of proof steps are listed in table 2 along with examples. Unparseable proof steps are marked as incorrect. For brevity, we refer to strictly-valid atomic correct steps as *canonical steps*. Psuedocode of the procedure to evaluate proofs is given in algorithm 1 in the Appendix.

**Metrics.** Given the above categorization of proof steps, a proof is defined to be correct if and only if there exists a path of proof steps from the premises to the conclusion (note that under this definition, it is possible for a correct proof to contain invalid proof steps). We could require that all proof steps in the path be canonical. But it is not obvious that this metric, which we call *strict proof accuracy*, would accurately measure the reasoning ability of the model. As such, we also consider more relaxed metrics: (a) we allow proof steps in the path to be strictly-valid non-atomic correct, which we call *"skip" proof accuracy*, (b) we allow proof steps to be broadly-valid, which we call *broad proof accuracy*, or (c) we allow proof steps to be strictly- or broadly-valid, which we call *valid proof accuracy*.

## 5 RESULTS

### 5.1 EXPERIMENTAL SETUP

In each experiment, we generate QA examples, perform CoT prompting on the LLMs, and analyze the predicted CoTs. We run the experiments on INSTRUCTGPT and the original GPT-3 (OpenAI

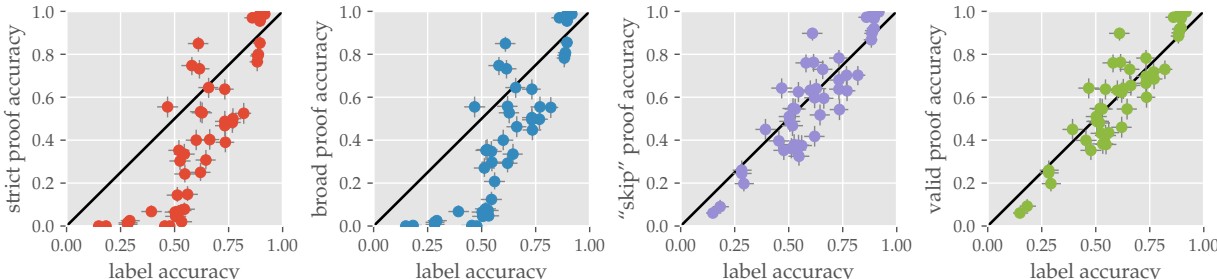

FIGURE 3: Scatter plots of label accuracy vs proof accuracy of all GPT-3 experiments in this paper. The black line indicates perfect agreement between label accuracy and proof accuracy. We emphasize that "proof accuracy" indicates the fraction of proofs (not proof steps) that are considered correct according to our metrics. Label accuracy is not well-correlated with strict or broad proof accuracy, and is better correlated with "skip" and valid proof accuracy, suggesting that label accuracy is a good measure of reasoning ability.

models `text-ada-001`, `text-babbage-001`, `text-curie-001`, `davinci`, `text-davinci-001`, `text-davinci-002`), with greedy decoding (Ouyang et al., 2022; Brown et al., 2020). We use 8-shot in-context learning, so each input to the LLM consists of 8 fully-labeled questions followed by a single test question with missing CoT and label. The model's task is to predict the CoT and label for the test question. Note that all examples across all inputs are independently and identically generated from PRONTOQA.

There are a number of variables that we control when generating examples in PRONTOQA: (1) the number of hops, (2) the ordering in which the sentences are generated from the ontology, and (3) the type of the ontology. The number of hops directly controls the difficulty of the generated example, and we experiment with 1, 3, and 5 hops.

We control the ontology traversal direction: We either traverse the tree top-down (i.e., preorder) or bottom-up (i.e., postorder), generating a sentence for each traversed edge/node. The ordering also affects the difficulty of the generated example: if the sentences are generated bottom-up, they will follow the same order as the steps in the gold proof. On the other hand, if they are generated top-down, the order is reversed, and the task may be more difficult.

To avoid any confounding effects from knowledge acquired during pretraining, PRONTOQA generates examples with fictional concept names (e.g., "wumpus" instead of "cat," etc). But we are also interested in measuring this confounding effect, and so in addition to fictional ontologies, we also generate "true" and "false" ontologies. True ontologies use real concept names and are consistent with the real-world (we randomly sample from a list of three hand-coded real ontologies). False ontologies use real concept names but the trees are generated using the random process described in section 3, and so it is very likely to generate a false statement, such as "All mammals are cats."

For each combination of variables, we run the model on 400 examples generated from the testbed, for a total of 48 experiments. We compute 95% confidence intervals for each experiment, as the number of correct proofs is distributed as $Binomial(400, p)$ with $p$ being the model's accuracy (Wilson, 1927).

## 5.2 DO CORRECT ANSWERS IMPLY CORRECT REASONING?

Label accuracy may not necessarily measure whether the model is performing reasoning correctly, since the model may find ways to guess the label via heuristics. To gauge whether label accuracy is a good metric and which proof accuracy metric is best to measure reasoning ability, we investigate how label accuracy is related to the various proof accuracy metrics. We plot proof accuracy vs label accuracy (i.e., simply checking whether the predicted label "True" or "False" is correct) of every experiment that we conducted in figure 3. Each point in the scatter plot corresponds to one of our 48 experiments described above. Observe that the label accuracy is poorly correlated with strict proof accuracy, and that strict proof accuracy may underestimate the model's reasoning ability. Rather, the most permissive accuracy metric has the highest correlation with label accuracy, suggesting that label accuracy is appropriate to measure reasoning accuracy. It also suggests that the most permissive proof accuracy metric is most appropriate for measuring the reasoning ability of the model.

## 5.3 PROOF ANALYSIS RESULTS

**Only the largest model is able to reason.** We investigated how reasoning ability is affected by model size. In figure 8 in the Appendix, proof accuracy increases considerably when increasing the model size from 350M to 1.3B and 6.7B. However, only `text-davinci-002` is able to perform

better than chance. We were not able to conclusively discern the cause of the significant difference in performance between version 001 and 002. One possible factor is the maximum token limit of version 002 is roughly twice that of version 001. In fact, the model `davinci` seems to perform as well as, if not slightly better than, `text-davinci-001`. In addition, we notice that the frequency of invalid steps decreases as the model size increases, and so larger models seem to be better at making valid steps, whether or not those steps are actually useful.

For the remainder of the paper, our results focus on `text-davinci-002`. Our main results are in figure 4 where we show the proof accuracy and distribution of proof step types for all experiments.

**Real-world knowledge helps reasoning.** We investigate the extent to which reasoning ability is affected by whether the ontology is fictional, "true," or "false." Evidently from figure 4, the LLM seems to perform comparably in the fictional and "false" ontology settings (accuracy is slightly worse with a "false" ontology). But when using the "true" ontology, the model performs much better, and its performance does not drop when increasing the number of hops from 3 to 5. The model is able to utilize its background knowledge from pretraining to "skip" hops, and is thus not as negatively affected by the increased hops. This is consistent with the findings of Dasgupta et al. (2022).

Evidently, the model's reasoning is heavily reliant on real-world knowledge, and this may be a problem for generalizability, such as when applying LLMs to novel scenarios or to settings that are not well-represented in the training data.

**Longer proofs are still challenging.** We investigate the extent to which reasoning ability is affected by the number of hops in the proof. We see from figure 4 that the model handles 1- and 3-hop examples quite well but struggles with 5-hop top-down examples, with accuracy falling to chance. So while it is able to perform reasoning to an extent, it is more limited as the number of hops increases.

**Traversal direction affects reasoning.** We also tested how reasoning ability is affected by the traversal direction of the ontology. We notice in figure 4 that as the number of hops increases, the model becomes sensitive to the traversal direction of the ontology (top-down vs bottom-up). This may be due to the fact that the order of the gold proof steps mirrors the bottom-up traversal, and is the reverse of the top-down traversal. Thus, the task may be made more difficult for language models if the context sentences are ordered top-down.

**How do LLMs reason step-by-step?** We investigate the fraction of correct and incorrect proofs that contain various types of proof steps, and whether the correctness of the proof is correlated with the presence of specific types of proof steps. Figure 4 breaks down the bars further (in darker red and blue) to indicate the fraction of proofs that contain proof steps other than canonical steps, since most predicted proof steps were canonical (in the 5-hop experiments with fictional ontology, they constitute $93.2\%$ of proof steps). We make the following observations:

1. Most predicted proof steps are strictly-valid (in the 5-hop experiments with fictional ontology, $93.2\%$ of proof steps are strictly-valid, $2.4\%$ are broadly-valid, and $5.9\%$ are invalid).
2. LLMs tend to skip steps by producing non-atomic steps, just as humans do when they verbalize their reasoning (in the 5-hop experiments with fictional ontology, $2.4\%$ of proof steps are non-atomic, even though all steps in the few-shot examples are atomic).
3. Most incorrect proofs contain misleading steps and invalid steps. This suggests that the source of the incorrect reasoning is either a due to a misleading step or an invalid step that causes the model to produce steps that do not belong to the gold proof.

Intriguingly, some correct proofs also contain misleading steps and invalid steps, which implies that the model is sometimes able to recover from these "mistakes" and return to the gold proof. We analyze this behavior in greater detail in section 5.4.

## 5.4 WHAT LEADS TO A MISTAKE?

We investigate whether specific types of proof steps are causing INSTRUCTGPT to produce reasoning errors. To do so, we identify the first step in each incorrect proof that is not a canonical step. We observe in figure 5, among incorrect proofs, strictly-valid atomic misleading steps appear in the proof first far more often than other non-canonical step types, including invalid steps. See figure 7 in the appendix for an example prediction where a misleading step causes the model to fail to prove the goal and produce an invalid step. This indicates that for the best-performing models, the main source of reasoning error is from misleading steps, since most predicted steps are strictly-valid and atomic. That is, imagining the space of proof steps as a graph where each edge represents a single valid step, INSTRUCTGPT almost always performs a walk in this graph. Once INSTRUCTGPT encounters a branch where one path at the fork follows the correct proof and the other paths do not,

FIGURE 4: Proof accuracy versus ontology type, number of hops, and ontology traversal direction. Each bar is subdivided into six darker bars according to the types of proof steps that appear in the predicted chains-of-thought. For example, the dark red bar corresponding to "invalid steps" indicates the proportion of incorrect proofs that contain an invalid step. The dark blue bar corresponding to "invalid steps" indicates the proportion of correct proofs that contain an invalid step. The proof step types are detailed in figure 2.

INSTRUCTGPT will select the incorrect direction with some frequency and is then not able to return to the correct path. Therefore, it seems that while LLMs are able to produce valid proof steps with high probability, they have difficulty with proof planning/strategizing.

We were curious if this relationship held in smaller models. We see in figure 5 that smaller models are more prone to make invalid or non-atomic steps as their first non-canonical step. But as model size increases, these types of steps become rarer, and is instead superseded by misleading steps.

Looking again at figure 4, we note that many correct proofs also contain misleading steps, and so it must be the case that INSTRUCTGPT sometimes returns to the correct proof path at some point after making a misleading step. To investigate this behavior more closely, we count the number of steps that the model takes *after* making a misleading step until it produces a step in the gold proof and plot the histogram in figure 10 in the appendix. We observe that, in general, the more time the model spends outside the correct proof path, the less likely it becomes to return to the correct proof.

We demonstrate in section A.7 in the appendix that our findings generalize to more sophisticated prompting strategies via an experiment using *self-consistency* prompting (Wang et al., 2022) and an experiment using a prompt containing example traces of depth-first proof search (i.e. containing examples of the search recovering from misleading steps).

FIGURE 5: Proportion of incorrect proofs versus the type of the first error (i.e., non-canonical proof step) and model size. The proof step types are detailed in figure 2. We note that in the 3-hop experiments with fictional ontology, four of the 400 examples surpassed the 2049 token limit for all models (except `text-davinci-002`). These examples were ignored (so the effective number of examples is 396). We omit the results for the 1-hop experiments here since there were too few incorrect proofs.

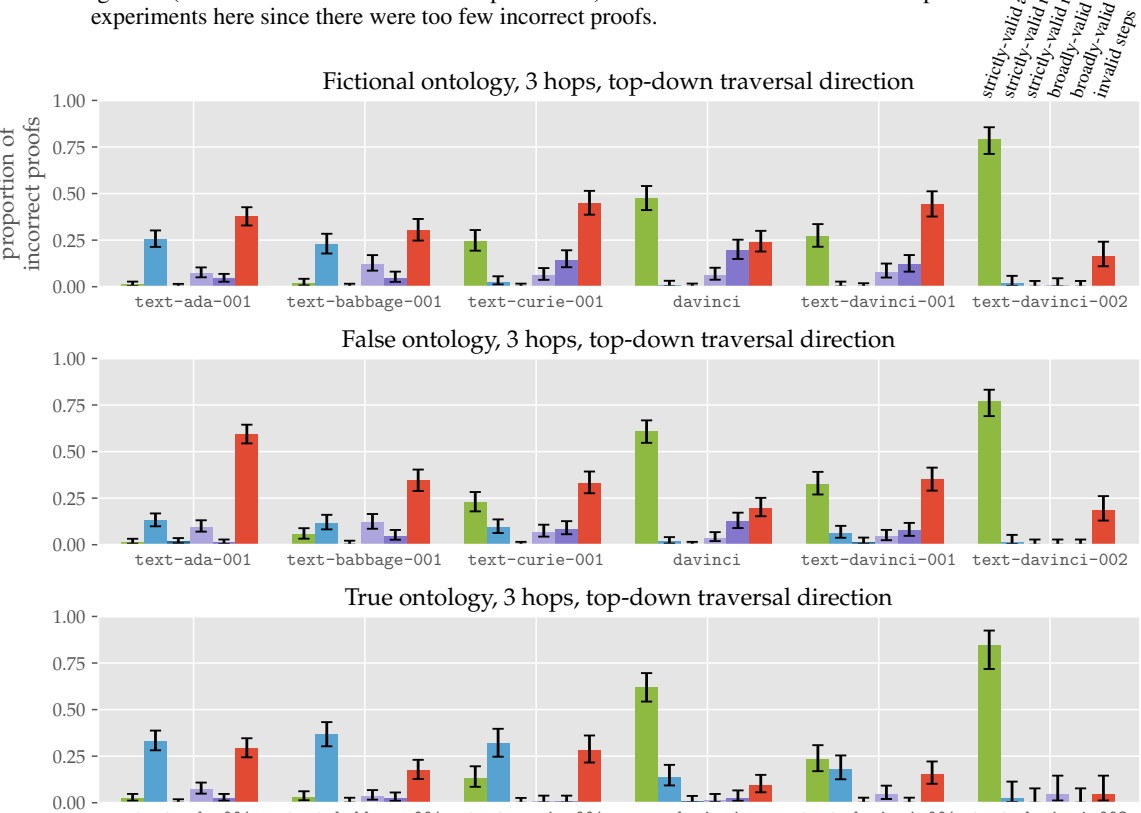

## 6 CONCLUSION AND FUTURE WORK

In this work, we introduced a synthetic fictional QA dataset called PRONTOQA designed to evaluate the reasoning ability of LLMs. We evaluated INSTRUCTGPT and GPT-3 on PRONTOQA and found that while the largest model was generally able to perform reasoning, it had difficulty with proof planning and selecting the correct proof step when there are multiple available.

PRONTOQA, and our high-level approach more broadly, could be used to compare LLM reasoning with that of humans, and to explore which aspects of human reasoning were acquired by LLMs from their pretraining. As our work has shown that LLMs are able to reason to a degree, it is yet unclear where the model acquired this ability. Are there portions of the pretraining data that teach the model to reason? Our work shows that CoT prompting is not sufficient for more complex reasoning, such as in mathematical domains, since the reasoning tested in this work is a strict subset of that of general mathematical reasoning. Mathematical proofs contain steps with much higher branching factor, where robust proof planning is instrumental. Rather, our results suggest that reasoning systems may benefit from more sophisticated proof planning/search strategies, such as neurosymbolic approaches where part of the reasoning is done over interpretable symbolic structures. PRONTOQA can be used to train new reasoning systems, or to pretrain/fine-tune LLMs to improve their reasoning capability.

The inability of LLMs to plan ahead in their reasoning might be related to recent work illuminating the theoretical computational limitations of such models (Merrill et al., 2022).

Since our analysis was limited to modus ponens, proof lengths of at most 5, and semantically simple sentences, it remains to be seen whether LLMs are able to produce longer proofs, or reason with other deduction rules, or over more semantically complex sentences/logical forms.

REPRODUCIBILITY STATEMENT

All our experiments in the main text were run using the OpenAI API on September $9^{th}$, $10^{th}$, and $11^{th}$, 2022. The self-consistency experiment was run on October $29^{th}$ and $30^{th}$, and the DFS experiment was run on November $16^{th}$ (see section A.7). For the sake of reproducibility of the analysis, all model outputs, the code for data generation, and the analysis code are freely available with a permissive open-source license at `github.com/asaparov/prontoqa`. The command `python analyze_results.py` produces all figures used in this paper.

ACKNOWLEDGMENTS

We thank Vishakh Padmakumar, Richard Yuanzhe Pang, Nitish Joshi, Daniel Khashabi, Nicholas Lourie, and Will Merrill for their helpful and insightful discussion. This research was supported by Open Philanthropy, Samsung Advanced Institute of Technology (Next Generation Deep Learning: From Pattern Recognition to AI), AWS AI, and Cisco Research.

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

## A APPENDIX

### A.1 DEDUCTION RULES

Figure 6 outlines the two deduction rules that we utilize in PRONTOQA.

| **Deduction rules in general form** | **Examples** |
|---|---|

$$\frac{f(a) \quad \forall x(f(x) \to g(x))}{g(a)} \text{ Hop} \qquad \frac{\text{cat}(\text{fae}) \quad \forall x(\text{cat}(x) \to \text{carnivore}(x))}{\text{carnivore}(\text{fae})} \text{ Hop}$$

*i.e., Given that "Fae is a cat" and "All cats are carnivores," we conclude that "Fae is a carnivore."*

$$\frac{}{A} \text{ Ax} \qquad \frac{}{\text{cat}(\text{fae})} \text{ Ax}$$

*i.e., Assume that "Fae is a cat" is an axiom.*

FIGURE 6: The two deduction rules that constitute the restricted proof calculus in our experiments. All proofs in PRONTOQA are composed of instances of only these two rules. Here, $A$ is any expression, and $f(a)$ is any expression where the variable $x$ in $f$ is substituted with any term $a$ (and similarly for $g(a)$).

### A.2 AVOIDING SHORTCUTS

When generating examples in PRONTOQA, we have to be careful to remove any shortcuts in the question that would allow the model to "guess" the answer without reasoning. In fact, we find that without any distractors, INSTRUCTGPT is able to predict the "true"/"false" label almost perfectly. INSTRUCTGPT can utilize the heuristic that whether the queried property is mentioned in the context implies whether or not it is true. For instance, if the example is asking "Sally is a cat. True or false: Sally is a vertebrate," the model can simply look for a string "Every ___ is (not) a vertebrate,"

regardless of the content in the blank. Due to the generative process of these examples, this kind of sentence is guaranteed to appear exactly once in the context. Thus, to ensure that such a heuristic is not informative, we add a distractor sentence by generating a novel concept that is disconnected from the ontology tree, and we assign to this new concept the negation property that is queried by the question. So in the above example, if the ontology has the rule "Every mammal is a vertebrate," a possible distractor sentence is "Every insect is not a vertebrate." We insert this distractor sentence into a random position in the context.

## A.3  EXAMPLE INSTRUCTGPT MISPREDICTION

```
Q: Every vumpus is a numpus.  Each vumpus is dull.  Dumpuses are vumpuses.
Every dumpus is not hot.  Every impus is a dumpus.  Impuses are brown.
Wumpuses are impuses.  Wumpuses are opaque.  Tumpuses are wumpuses.  Tumpuses
are small.  Every yumpus is hot.  Zumpuses are tumpuses.  Zumpuses are feisty.
Rompuses are zumpuses.  Every rompus is not kind.  Each jompus is a rompus.
Jompuses are sour.  Alex is a zumpus.  True or false:  Alex is hot.
A:
```

**Expected answer:** Alex is a zumpus. Zumpuses are tumpuses. Alex is a tumpus. Tumpuses are wumpuses. Alex is a wumpus. Wumpuses are impuses. Alex is an impus. Every impus is a dumpus. Alex is a dumpus. Every dumpus is not hot. Alex is not hot. False

**Predicted answer:** Alex is a zumpus. Zumpuses are tumpuses. Alex is a tumpus. Tumpuses are wumpuses. Alex is a wumpus. Wumpuses are impuses. Alex is an impus. Every impus is a dumpus. Alex is a dumpus. Dumpuses are vumpuses. Alex is a vumpus. Every vumpus is a numpus. Alex is a numpus. Numpuses are jompuses. Alex is a jompus. Jompuses are rompuses. Alex is a rompus. Rompuses are zumpuses. Alex is a zumpus. Zumpuses are feisty. Alex is feisty. True

FIGURE 7: An example from PRONTOQA (5 hops with fictional ontology and top-down traversal direction) along with the expected and predicted answer from INSTRUCTGPT. Note that most of the predicted steps are canonical (highlighted yellow). The model makes a single misleading step (highlighted purple) which causes it fail to prove the goal, and to eventually make an invalid step (highlighted red).

A.4 HOW WE EVALUATE THE CHAIN-OF-THOUGHT

---

**Algorithm 1:** Our algorithm for reconstructing and evaluating the proof from the predicted chain-of-thought, and for computing whether each proof step is valid vs invalid, atomic vs non-atomic, misleading vs correct. Here, we use the notation $\varphi[x \mapsto c]$ to denote the substitution of all occurrences of the symbol $x$ with $c$ in the logical form $\varphi$. We use the helper function `is_provable` to compute whether a given logical form $\varphi$ is provable from a set of axioms with one or more deduction rules. The function returns a tuple $(P, k)$ where if $k \geq 0$, $\varphi$ is provable in $k$ steps using the premises $P$. Otherwise, $\varphi$ is not provable.

---

```
1  function evaluate_cot(context sentences Q₁,...,Qₘ,
                         predicted chain-of-thought sentences C₁,...,Cₙ,
                         gold chain-of-thought sentences T₁,...,Tᵣ)
2      for i ∈ 1,...,m do                              /* parse the context */
3        ⌊ Lᵢ^Q = semantic_parse(Qᵢ)
4      for i ∈ 1,...,r do                      /* parse the gold chain-of-thought */
5        ⌊ Lᵢ^T = semantic_parse(Tᵢ)
6      initialize S as an empty set
7      for i ∈ 1,...,n do          /* parse and evaluate the predicted chain-of-thought */
8        ⌈ Lᵢ^C = semantic_parse(Cᵢ)
9        | (P,k) = is_provable(Lᵢ^C, {L₁^Q,...,Lₘ^Q}, S)
10       | if k ≥ 0
         |   ⌈ /* if we wish to use a stricter metric for proof accuracy, we can add
         |   |    conditions here (e.g., requiring atomicity by checking k = 1) */
11       |   ⌊ add Lᵢ^C to S
12       | if P ⊆ {L₁^T,...,Lᵣ^T} and Lᵢ^C ∉ {L₁^T,...,Lᵣ^T}
         |   ⌈ /* the premises are in the gold proof but the conclusion is not */
13       |   ⌊ mark Lᵢ^C as a misleading step

14     return Lᵣ^T ∈ S       /* the proof is correct if the final conclusion is provable */
15  function is_provable(logical form φ, set of axioms A, previous conclusions S)
16     if φ ∈ A
17       | return ({φ}, 1)                          /* provable by Ax step (strictly-valid) */
18     else if φ ∈ S
19       | return ({φ}, 0)                          /* already proved by previous step */
20     else if φ has form g(c) or ¬g(c) for any constants g and c
21       | for a ∈ A ∪ S do
22       |   | if a has form ∀x(ψ → γ) where γ[x ↦ c] = φ
23       |   |   | (P,k) = is_provable(ψ[x ↦ c], A, S)
24       |   |   | if k ≥ 0
25       |   |   |   ⌊ return (P ∪ {a}, k + 𝟙{a ∈ A})    /* provable by Hop step (strictly-valid) */

26     else if φ has form ∀x(ψ → γ)
         /* note:  we precompute this graph */
27       | let G be the graph where for any axiom in A with form ∀x(α → β), α and β are vertices and there is a
         |   directed edge from α to β
28       | if there is a path in G from ψ to γ
         |   ⌈ /* provable with additional deduction rules (broadly-valid) */
29       |   ⌊ return ( axioms corresponding to path edges , length of path )

30     return (∅, -1)                          /* this step is not provable (i.e., invalid) */
```

---

## A.5 PROOF ACCURACY VS MODEL SIZE

FIGURE 8: Proof accuracy versus model size, ontology type, and number of hops. Each bar is subdivided into six bars according to the types of proof steps that appear in the predicted chains-of-thought. The proof step types are detailed in figure 2. Top-down traversal direction is used in these experiments. We note that in the 3-hop experiments with fictional ontology, four of the 400 examples surpassed the 2049 token limit for all models (except `text-davinci-002`). These examples were ignored (so the effective number of examples is 396).

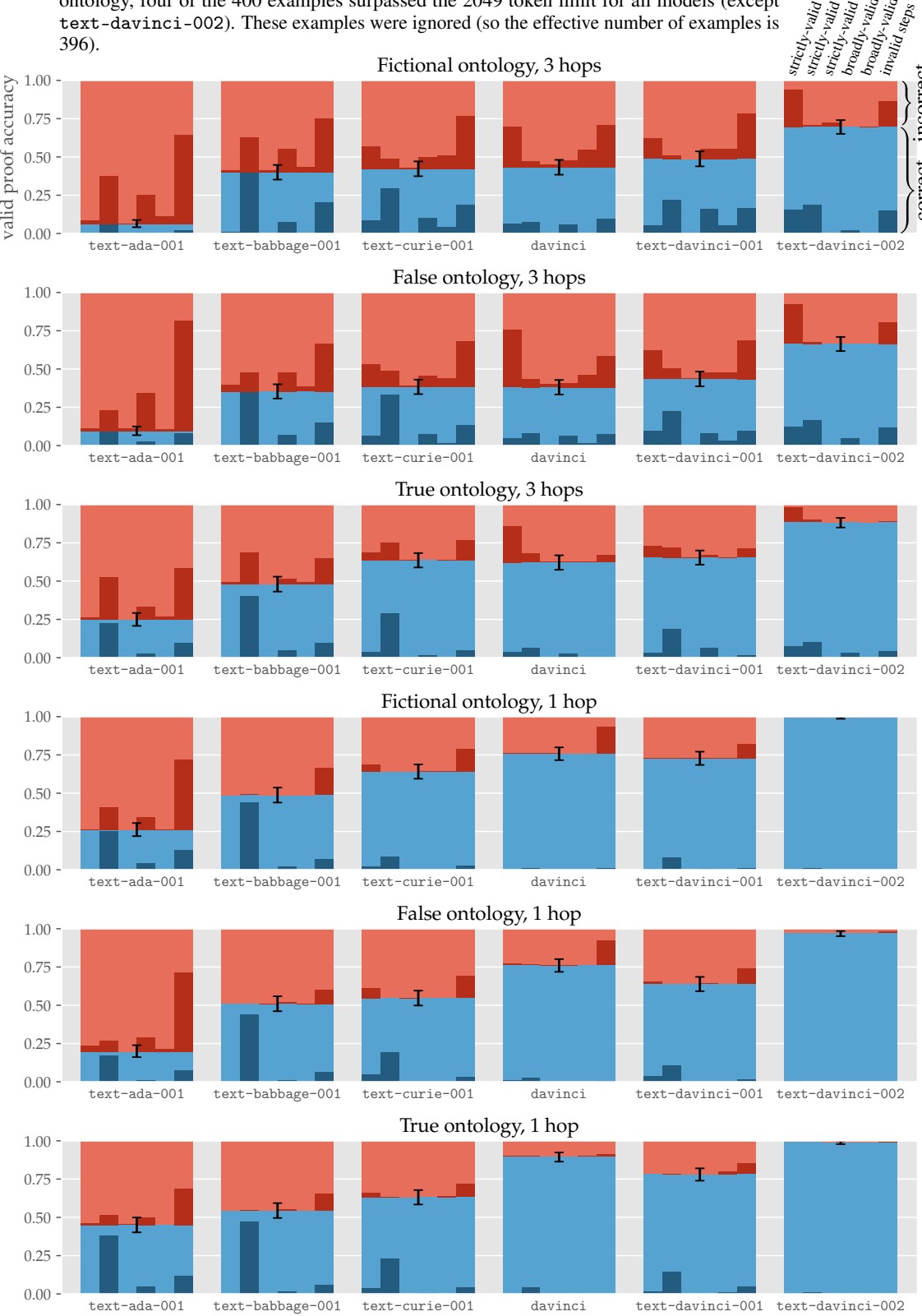

## A.6   ADDITIONAL ERROR ANALYSIS

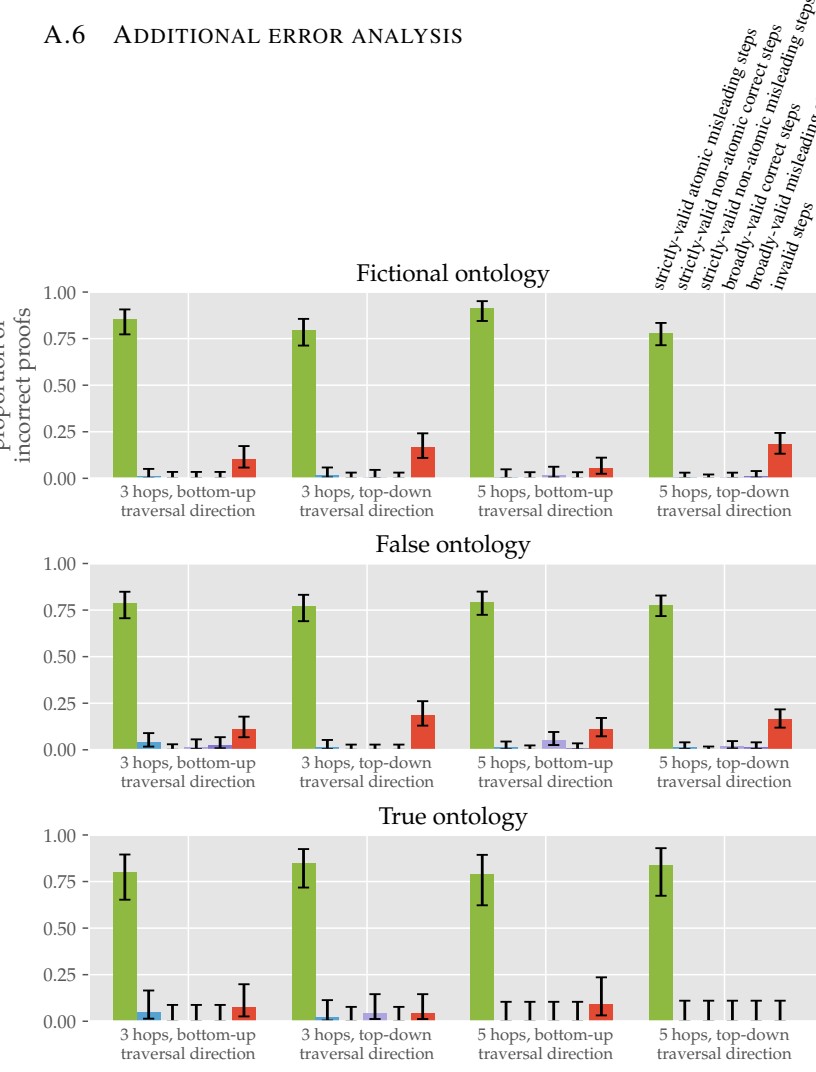

FIGURE 9: Proportion of incorrect proofs versus the type of the first error (i.e., non-canonical proof step), number of hops, and ontology traversal direction. The proof step types are detailed in figure 2. We note that in the 3-hop experiments with fictional ontology, four of the 400 examples surpassed the 2049 token limit for all models (except `text-davinci-002`). These examples were ignored (so the effective number of examples is 396). We omit the results for the 1-hop experiments here since there were too few incorrect proofs.

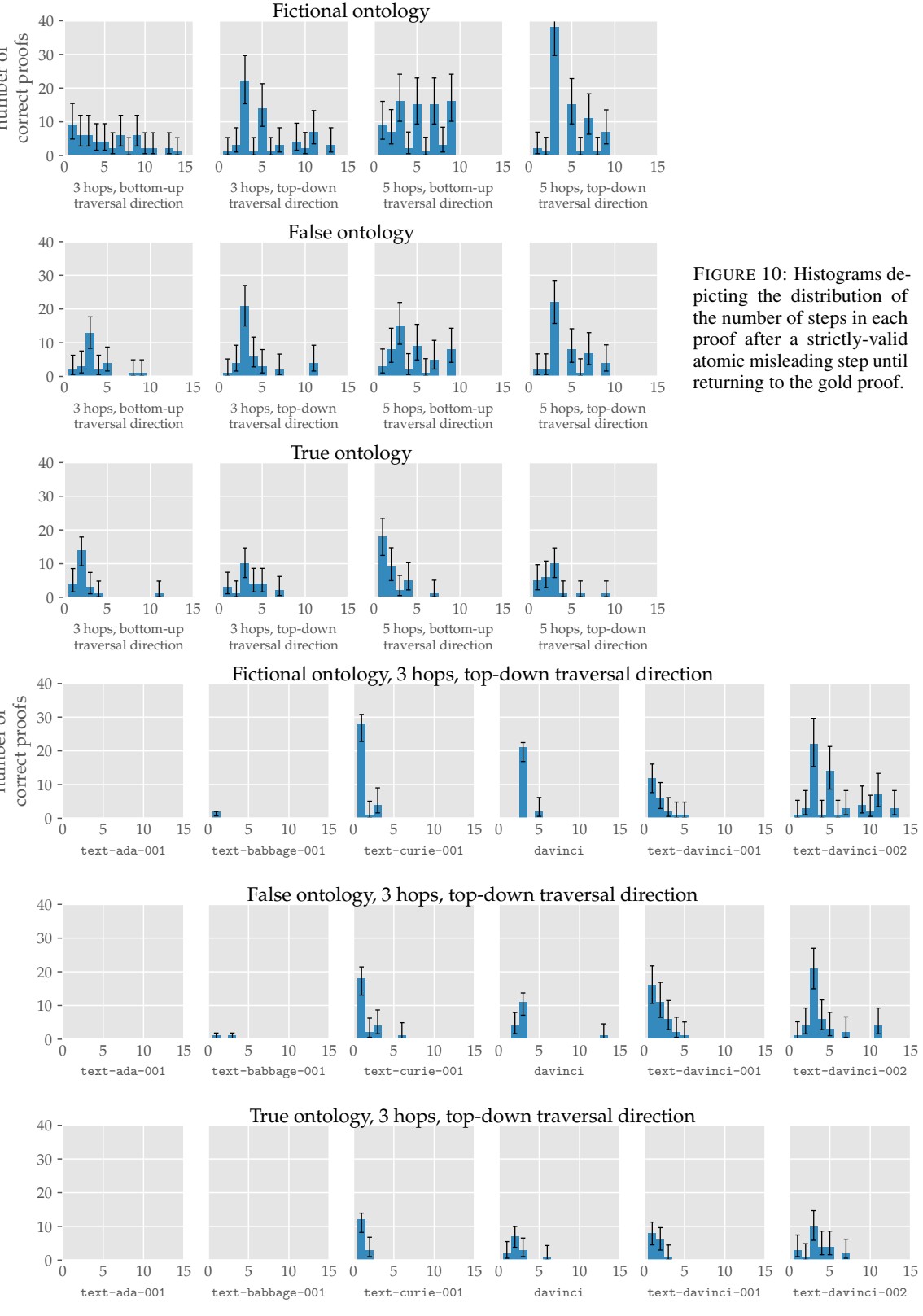

FIGURE 10: Histograms depicting the distribution of the number of steps in each proof after a strictly-valid atomic misleading step until returning to the gold proof.

### A.7    DO OTHER PROMPTING STRATEGIES HELP?

### A.7.1    SELF-CONSISTENCY

To what extent do our findings generalize to prompting strategies other than CoT with greedy decoding? To test this, we experimented with *self-consistency* prompting (Wang et al., 2022), where for each example, we queried the LLM for $40$ sample predictions of the CoT, using a temperature setting of $0.7$. For each sample CoT $s_i$, we compute the following quantity:

$$\exp\left\{\frac{1}{|s_i|}\sum_{j=1}^{|s_i|}\log p(s_{i,j}|s_{i,1},\ldots,s_{i,j-1})\right\}.$$

We parse each predicted CoT sample into a sequence of logical forms, and we find the logical form sequence with the highest sum of the above quantity over all the CoT samples that share the same semantic parse. This logical form sequence is the final prediction.

We run this experiment in our setting with 5 hops, fictional ontology, and top-down traversal direction, with $100$ examples. The resulting valid proof accuracy is $0.56$ compared to $0.545$ which is not significantly different. Furthermore, inspecting specific examples of CoT samples (see figure 11), we see that for examples that the model gets wrong, the model is actually assigning higher overall probability to the incorrect proof than to the correct proof. This suggests that our results do in fact generalize to more sophisticated prompting/decoding strategies, and that strategies that endeavor to find proofs with higher probability globally (e.g. beam search) will not help the model in proof planning.

### A.7.2    CAN THE MODEL LEARN TO DO DEPTH-FIRST SEARCH FROM IN-CONTEXT EXAMPLES?

Our earlier analysis also revealed a possible way forward to rectify the model's shortcoming in proof planning: Even after making a misleading step, INSTRUCTGPT sometimes "returned" to the correct proof. We could instead relax the constraint that the chains-of-thought always reflect the shortest correct proof of the answer. Instead, we allow the in-context examples to contain misleading steps, with the hope that the model learns to better recover from misleading steps. It is reasonable for humans to explore a space of possible solutions before arriving at the correct answer, and we could mimic this in LLMs by allowing the CoT to explore alternative paths in the space of proof steps, even if those paths are not ultimately part of the shortest proof. This kind of search strategy is analogous to depth-first search (DFS) in graphs.

To test whether this approach improves the model's reasoning ability, we conduct an experiment where we provide in-context examples of chains-of-thought that follow a DFS, with the hope that the model is able to learn to perform DFS when given a new test example, thereby improving the likelihood that it finds the correct answer. We run this experiment in our setting with 5 hops, fictional ontology, and top-down traversal direction, with 100 examples. The resulting valid proof accuracy is $0.55$ compared to $0.545$ which, again, is not significantly different.

```
Q: Each rompus is a wumpus.  Every rompus is not opaque.  Every jompus is a
rompus.  Every jompus is not sour.  Vumpuses are jompuses.  Each vumpus is
not earthy.  Yumpuses are vumpuses.  Each yumpus is angry.  Each tumpus is a
yumpus.  Tumpuses are large.  Numpuses are tumpuses.  Numpuses are not cold.
Every zumpus is a numpus.  Dumpuses are not angry.  Zumpuses are shy.  Impuses
are zumpuses.  Impuses are luminous.  Polly is an impus.  True or false:  Polly
is angry.
A:
```

**Expected answer:** Polly is an impus.  Impuses are zumpuses.  Polly is a zumpus.
Every zumpus is a numpus.  Polly is a numpus.  Numpuses are tumpuses.  Polly is
a tumpus.  Each tumpus is a yumpus.  Polly is a yumpus.  Each yumpus is angry.
Polly is angry.  True

**Sample predicted answer:** Polly is an impus. Impuses are zumpuses. Polly is a zumpus. Zumpuses are numpuses. Polly is a numpus. Numpuses are tumpuses. Polly is a tumpus. Every tumpus is a yumpus. Polly is a yumpus. Yumpuses are vumpuses. Polly is a vumpus. Vumpuses are jompuses. Polly is a jompus. Each jompus is a rompus. Polly is a rompus. Rompuses are wumpuses. Polly is a wumpus. Wumpuses are not angry. Polly is not angry. False
**Sample average log probability:** −0.07272354467015028

. . .

**Sample predicted answer:** Polly is an impus. Impuses are zumpuses. Polly is a zumpus. Zumpuses are numpuses. Polly is a numpus. Numpuses are tumpuses. Polly is a tumpus. Each tumpus is a yumpus. Polly is a yumpus. Yumpuses are angry. Polly is angry. True
**Sample average log probability:** −0.04729328860082488

**Sample predicted answer:** Polly is an impus. Impuses are zumpuses. Polly is a zumpus. Zumpuses are numpuses. Polly is a numpus. Numpuses are tumpuses. Polly is a tumpus. Each tumpus is a yumpus. Polly is a yumpus. Yumpuses are vumpuses. Polly is a vumpus. Vumpuses are jompuses. Polly is a jompus. Every jompus is a rompus. Polly is a rompus. Rompuses are not angry. Polly is not angry. False
**Sample average log probability:** −0.02227105943358164

. . .

FIGURE 11: An example from PRONTOQA (5 hops with fictional ontology and top-down traversal direction) using self-consistency (each sample was produced using a temperature of 0.7), showing the expected and sample predicted answers from INSTRUCTGPT. Canonical steps are highlighted yellow, misleading steps purple, and invalid steps red. We note that the sample predicted CoTs that correspond to the gold proof are given lower overall probability than those that are incorrect.

