# OpenReview forum: "Language Models Are Greedy Reasoners: A Systematic Formal Analysis of Chain-of-Thought"
_ICLR.cc/2023/Conference — ICLR 2023 poster_

### Official Review · Reviewer_UrF5 · 2022-10-13

**Confidence:** 5
**Correctness:** 3
**Technical Novelty And Significance:** 3
**Empirical Novelty And Significance:** 3
**Recommendation:** 8

**Clarity, Quality, Novelty And Reproducibility:**

Novelty:

1. The proposed dataset is exactly the same as that presented in ProofWriter simply with differently named predicated. There is no direct comparison to ProofWriter and ProofWriter could have been used to perform these experiments. Proof accuracy was reported in ProofWriter and also [2].

2.  “We use a simple grammar to convert the formal statements of the ontology into the natural language utterances that make up the context.”  The authors should reference [1] since they use similar templates. I would also argue that this is still a synthetic language, not a natural language.

Quality, Clarity and Reproducibility:

3. It is great that the authors purposefully force the model to avoid shortcuts. However, if you “add a distractor sentence by generating a novel concept that is disconnected from the ontology tree” the model could just learn to ignore the rules whose concepts do not come from the ontology.

4. “To achieve this, we parse each sentence of the predicted CoT into logical form” - How is this done? What if the samples are not in the correct format?


5. Why do the authors claim “it is not obvious that this metric, which we call strict proof accuracy, would accurately measure the reasoning ability of the model.”? For reasoning, strict proof accuracy may be very important.

6. While, results in Figure 3 are very powerful! However, it would also be great to have some indication of depth of the reasoning problems. For example, I would imagine for depth-1 reasoning problems the correlation would be very strong and possibly less so for problems of depth-5.

7. It is interesting (if not unsurprising) in Figure 4 that the model appears to be dependant on its priors (i.e. best results with the True Ontology).

8. To make Figure 5 more easy to parse it may be a good idea to plot these as a pie char showing that they are making up some proportion of the error and include in each caption the original proportion of incorrect predictions.

9. The sub-title “Reasoning is an emergent ability.” Is very misleading. Please rephrase this.

10. For completeness please provide the prompt used for CoT and also include some example samples in the appendix.

11. The authors only show results for a single model. It would be best to evaluate other models too.

Refs:
[1] Critical Thinking Thinking for Language Models, Betz et al. 2020 \
[2] Faithful Reasoning Using Large Language Models, Creswell et al, 2022

**Strength And Weaknesses:**

Strengths:
1. It’s great that the authors evaluate both validity, atomicity and wether a step is misleading. These are very important metrics.
2. Results in Figure 3 are very powerful!
3. The three observations on page 8 are very informative.
4. If the analysis code is made available, I can imagine this being a very useful tool for people to measure reasoning progress. However, this would be better if it had leveraged existing datasets e.g. ProofWriter.

Weaknesses:
1. The authors ignore a significant chunk of prior work.
2. They propose a novel dataset, while there already exists a dataset, ProofWriter, that would have been suitable for these experiments and work by Betz et al. 2020 is also relevant (see detailed comments below).
3. The title is somewhat miss-leading. I would suggest only using the second half of the title.
4. Some of the claims are not supported, "Reasoning is an emergent ability".
5. Authors only show results for a single model.


**Summary Of The Paper:**

The authors propose the dataset, PRONTOQA, a reasoning dataset to evaluation Language model's reasoning capabilities. The authors provide an in-depth analysis of reasoning traces.

**Summary Of The Review:**

Recently there has been a lot of interested in language models (LMs) and their capabilities. Reasoning has emerged as an area of increasing interest and it's important that any capabilities are not over- (or under-) estimated. This paper offers many insights into the failings of Chain-Of-Thought prompting for LMs which I consider to be very valuable to the field. However, the (first half of the) title of the paper is misleading and the authors do not use a benchmark dataset. There are also a few points that need clarifying.

I am willing to recommend acceptance of the paper, conditional on the points above being satisfactorily addressed.

---

> ### Author Response · Authors · 2022-11-15
> **Response to Reviewer UrF5**
>
> Thank you for the feedback and helpful comments! We would greatly appreciate if the reviewer would let us know if some of the revisions were insufficient and we can further iterate on the paper during the revision period.
>
> 1. “The proposed dataset is exactly the same as that presented in ProofWriter simply with differently named predicated. There is no direct comparison to ProofWriter and ProofWriter could have been used to perform these experiments. Proof accuracy was reported in ProofWriter and also [Creswell et al. 2022].”
>
>   We also acknowledge the similarity of the proposed dataset PrOntoQA with ProofWriter, and we agree that a deeper comparison is warranted in the paper. We added a paragraph and a table at the beginning of the Related Work section to provide this comparison, but to expand further here: There are a number of reasons why we chose to create a new dataset rather than perform our analysis on ProofWriter.
>   - While the ProofWriter data is freely available, their code to generate the data is not, and our exploration benefited from the flexibility and controllability that is afforded by working with our own dataset, such as the ability to add distractors to mitigate heuristics.
>   - Each proof in ProofWriter contains a mix of implication and conjunction, as well as a mix of unary and binary predicates (e.g. “x is a cat” and “x is warm-blooded” are unary predicates whereas “x likes y” and “x has y” are binary). We endeavored to perform a more focused study where we would be able to measure differences in the model’s ability to reason over implication vs conjunction, or over binary vs unary predicates.
>   - One of the main efforts in our analysis is to measure the confounding effect of knowledge acquired from pretraining during reasoning, and we explicitly added a variable in our data generation to produce examples that were either consistent with the real world (true), inconsistent (false), or neither (fictional). However, ProofWriter does not make this distinction, with examples containing all three types of rules.
>
> 2. The reviewer brought missing related work to our attention.
>
>   We thank the reviewer for doing so! We have added a number of references to the Related Work section, including:
>   - Gregor Betz. Critical thinking for language models. arXiv, 2020.
>   - Antonia Creswell and Murray Shanahan. Faithful reasoning using large language models. arXiv, 2022.
>
> 3. “The title is somewhat misleading. I would suggest only using the second half of the title.”
>
>   We agree and have changed the title to “Language Models are Greedy Reasoners: A Systematic Formal Analysis of Chain-of-Thought” since we think it better characterizes our findings.
>
> 4. The claim that "reasoning is an emergent ability" is not supported.
>
>   We agree with the reviewer and have tempered this claim. In Section 5.3, we have changed the paragraph title to “Only the largest model is able to reason.”
>
> 5. The authors only show results for a single model. It would be best to evaluate other models too.
>
>   The reviewer may be referring to GPT-3 broadly, but we applied our analysis to a handful of models with varying scales. To help mitigate this potential ambiguity, we edited the footnote on page 2:
>   “Throughout the paper, InstructGPT refers to the model named text-davinci-002. But note that in our experiments, we also evaluate text-ada-001, text-babbage-001, text-curie-001, davinci, and text-davinci-001.”
>
> 6. “If you ‘add a distractor sentence by generating a novel concept that is disconnected from the ontology tree’ the model could just learn to ignore the rules whose concepts do not come from the ontology.”
>
>   This is true and this would be an example of another (slightly more complex) heuristic that the model could exploit. And this is a major motivation for our efforts to analyze the CoT itself rather than the predicted label, as we are able to determine whether the model actually reasoned over the ontology to arrive at the correct answer. Without the distractor, the label accuracy and proof accuracy would be less correlated. With the distractor, they are more correlated, but the distractor itself is not essential to our later results analyzing the CoT directly.
>
> 7. “‘To achieve this, we parse each sentence of the predicted CoT into logical form’ - How is this done? What if the samples are not in the correct format?”
>
>   We edited the following sentence in the second paragraph of section 4: “[We] parse each sentence of the predicted CoT into logical form via recursive-descent parsing using the simple grammar from the generative process.”
>   We also added the following sentence in the same section: “Unparseable proof steps are marked as incorrect.”
>   Interestingly, we only observed this to happen with the smaller models. The larger models always produced parseable CoT sentences.

---

> > ### Author Response · Authors · 2022-11-15
> > **Response to Reviewer UrF5 (continued)**
> >
> > 8. “Why do the authors claim ‘it is not obvious that this metric, which we call strict proof accuracy, would accurately measure the reasoning ability of the model.’? For reasoning, strict proof accuracy may be very important.”
> >
> >   It is true that strict proof accuracy can be very important in certain applications, such as mathematical problems where the task may require that every step be explicitly shown. Before seeing our results, it may also have been the case that strict proof accuracy is the most appropriate metric for evaluating the model’s performance on our reasoning task, which is why we considered it alongside the other metrics for proof accuracy in our experiment. We found that the model is able to arrive at the correct answer even if it produces non-atomic or broadly-valid steps, and therefore, strict proof accuracy tends to underestimate the reasoning ability of the model.
> >
> > 9. “For completeness please provide the prompt used for CoT and also include some example samples in the appendix.”
> >
> >   Thank you for the suggestion. We have added sections B and F to the appendix containing examples of questions, expected answers, and predicted answers from InstructGPT.

---

> > > ### Comment · Reviewer_UrF5 · 2022-11-21
> > > **Thank you for the changes, I'm happy to maintain my score.**
> > >
> > > - I appreciate that the authors need access to a dataset where they have more control and flexibility and that ProofWriter may not provide this.
> > > - I'm happy that the authors have changed the title of the paper and renamed the miss-leading section.

---

### Official Review · Reviewer_gxpW · 2022-10-24

**Confidence:** 4
**Correctness:** 3
**Technical Novelty And Significance:** 2
**Empirical Novelty And Significance:** 2
**Recommendation:** 6

**Clarity, Quality, Novelty And Reproducibility:**

The paper is clear and well written. There is some novelty in the new dataset, and in the results presented.

**Strength And Weaknesses:**

The authors’ dataset is quite similar to the ProofWriter dataset released by Allen AI. But the breakdown into fictional, false and true ontologies is novel and useful, and provides insight into the models’ capabilities.

I have some questions and thoughts.

The authors write: “the most permissive accuracy metric has the highest correlation with label accuracy, suggesting that GPT-3 is indeed performing the reasoning to answer the question, rather than relying on heuristics” (p.6). I don’t see how this conclusion follows. Isn’t this assuming causation from correlation? Also, since “valid proof accuracy” is strictly weaker than “strict proof accuracy”, isn’t it inevitable that the points in the scatter plot below the perfect line will be higher in the former compared to the latter? Wouldn’t this be the case however the model arrived at the answer?

The authors write: “Once InstructGPT encounters a branch where one path at the fork follows the correct proof and the other paths do not, InstructGPT will select the incorrect direction with some frequency and is then not able to return to the correct path. Therefore, it seems that while LLMs are able to produce valid proof steps with high probability, they have difficulty with proof planning/strategizing.” (pp.8-9). This account feels highly speculative. As the authors obviously know, LLMs like GPT don’t work like this at a fundamental level. They are just trying, autoregressively, to predict the next token in the given context. There is no explicit mechanism for “selecting” a branch in a proof tree, let alone for planning or strategizing, so it’s hardly surprising that they struggle with tasks where these things are required (because they are out of distribution). Maybe some cautionary remarks would be useful here.

The authors write: “we can extend the CoT paradigm where after every predicted step in the CoT, we perform beam search decoding to find the top-k most likely values for the next step. We can then perform beam search over the proof steps (i.e., sentences) rather than over the tokens.” (p.9). This has recently been done. See: A.Creswell & M.Shanahan, Faithful Reasoning Using Large Language Models, https://arxiv.org/abs/2208.14271.

**Summary Of The Paper:**

This paper presents a new reasoning dataset for LLMs, and uses it to assess the logical reasoning capabilities of a number of GPT variants, using chain-of-thought prompting, while varying a number of parameters. Using this dataset, the authors show expose a number of these models’ limitations, including their poor performance on reasoning tasks requiring multiple steps.

**Summary Of The Review:**

The new reasoning dataset offers some new features compared to similar existing datasets such as ProofWriter, and the paper reports a worthwhile investigation into LLM capabilities. However, the dataset is only a marginal improvement on ProofWriter IMO, and the results are unsurprising. So I don’t feel the paper offers a substantial enough contribution to merit acceptance at ICRL, given the low acceptance rate of the conference. It’s good work though, and could be a cornerstone of a larger research effort (e.g. improving the reasoning capabilities of LLMs), so the authors should be encouraged.

---

> ### Author Response · Authors · 2022-11-15
> **Response to Reviewer gxpW**
>
> Thank you for the feedback and helpful comments! We would greatly appreciate if the reviewer would let us know if some of the revisions were insufficient and we can further iterate on the paper during the revision period.
>
> 1. “[The] dataset is only a marginal improvement on ProofWriter…”
>
>   One important point that we wish to emphasize is that the main contribution of our work is not the dataset, but the analysis that is facilitated by the dataset, where we are able to formally inspect and study the predicted chains-of-thought of language models. Our analysis generalizes to other simple datasets, including ProofWriter. The analysis code is open source and freely available to anyone who wishes to apply it to their own data. To stress this point for readers, we add the following sentence to the second paragraph in the Introduction:
>   “We emphasize here that while the dataset is an important contribution of this paper, the main contribution is the analysis that is facilitated by the dataset.”
>
>   We also acknowledge the similarity of the proposed dataset PrOntoQA with ProofWriter, and we agree that a deeper comparison is warranted in the paper. We added a paragraph and a table at the beginning of the Related Work section to provide this comparison, but to expand further here: There are a number of reasons why we chose to create a new dataset rather than perform our analysis on ProofWriter.
>   - While the ProofWriter data is freely available, their code to generate the data is not, and our exploration benefited from the flexibility and controllability that is afforded by working with our own dataset, such as the ability to add distractors to mitigate heuristics.
>   - Each proof in ProofWriter contains a mix of implication and conjunction, as well as a mix of unary and binary predicates (e.g. “x is a cat” and “x is warm-blooded” are unary predicates whereas “x likes y” and “x has y” are binary). We endeavored to perform a more focused study where we would be able to measure differences in the model’s ability to reason over implication vs conjunction, or over binary vs unary predicates.
>   - One of the main efforts in our analysis is to measure the confounding effect of knowledge acquired from pretraining during reasoning, and we explicitly added a variable in our data generation to produce examples that were either consistent with the real world (true), inconsistent (false), or neither (fictional). However, ProofWriter does not make this distinction, with examples containing all three types of rules.
>   These concerns apply to FOLIO as well, but in addition, since FOLIO is human-annotated, we are not able to modify the data generation process to remedy the above issues.
>
> 2. “This account feels highly speculative. As the authors obviously know, LLMs like GPT don’t work like this at a fundamental level. They are just trying, autoregressively, to predict the next token in the given context. There is no explicit mechanism for “selecting” a branch in a proof tree, let alone for planning or strategizing, so it’s hardly surprising that they struggle with tasks where these things are required (because they are out of distribution).”
>
>   We do not assume a priori whether or not LLMs are able to reason, and we endeavor to study their abilities as a black box. LLMs have shown impressive out-of-distribution generalization results on many tasks related to reasoning [1, 2].
>   It is not unreasonable that a sufficiently expressive model with sufficient training data and compute would be able to further reduce training loss by more closely modeling the generative process of the observed text it strives to model. If the generative process of those observations requires reasoning to some degree (i.e. it would be computationally more difficult to model the data otherwise), then it bears examination that the language model might acquire some of this reasoning capability. It is very possible that InstructGPT has encountered some examples of these kinds of observations in their training, given its size. We believe it is worth the research effort to determine whether or not this is the case, to determine the nature of the reasoning capability if it exists, and to more closely examine large language models and their capabilities with a discerning eye, especially given the significant interest and investment by the community and industry.
>
> [1] Cem Anil, Yuhuai Wu, Anders Andreassen, Aitor Lewkowycz, Vedant Misra, Vinay Ramasesh, Ambrose Slone, Guy Gur-Ari, Ethan Dyer, and Behnam Neyshabur. Exploring Length Generalization in Large Language Models. arXiv, 2022.
> [2] Jason Wei, Xuezhi Wang, Dale Schuurmans, Maarten Bosma, Brian Ichter, Fei Xia, Ed Chi, Quoc Le, and Denny Zhou. Chain of Thought Prompting Elicits Reasoning in Large Language Models. arXiv, 2022.

---

> > ### Author Response · Authors · 2022-11-15
> > **Response to Reviewer gxpW (continued)**
> >
> > 3. The reviewer raises concerns with our argument: “The most permissive accuracy metric has the highest correlation with label accuracy, suggesting that GPT-3 is indeed performing the reasoning to answer the question, rather than relying on heuristics” (p.6).
> >
> >   We agree with the reviewer that our results do not prove causation, and so we have rephrased the sentence:
> >   “The most permissive accuracy metric has the highest correlation with label accuracy, suggesting that label accuracy is an appropriate way to measure reasoning accuracy.”
> >
> > 4. “Since ’valid proof accuracy’ is strictly weaker than ‘strict proof accuracy’, isn’t it inevitable that the points in the scatter plot below the perfect line will be higher in the former compared to the latter? Wouldn’t this be the case however the model arrived at the answer?
> >
> >   This is true, but the degree to which the points would move higher in the plot is not obvious a priori. For example, if the model were not correctly applying non-atomic or broadly-valid deduction steps, or if such steps were very rare, the points would barely move.
> >
> > 5. “The authors write: “we can extend the CoT paradigm where after every predicted step in the CoT, we perform beam search decoding to find the top-k most likely values for the next step. We can then perform beam search over the proof steps (i.e., sentences) rather than over the tokens.” (p.9). This has recently been done. See: [Creswell and Shanahan 2022].
> >
> >   We thank the reviewer for bringing this to our attention. We removed those sentences from the Conclusion and added section A.7 to the appendix describing experiments to test whether our results generalize to self-consistency prompting [3] and depth-first search. We found that neither self-consistency nor DFS improves proof accuracy on this task. We also find that for examples that the model gets wrong, the model actually assigns higher overall probability to the incorrect proof than to the correct proof. This suggests that prompting/decoding strategies that encourage the model to produce more globally-optimal outputs (e.g. beam search) would not remedy the model’s shortcomings in proof planning.
> >
> >   [3] Xuezhi Wang, Jason Wei, Dale Schuurmans, Quoc V. Le, Ed H. Chi, and Denny Zhou. Self-
> > consistency improves chain of thought reasoning in language models. arXiv, 2022.

---

> > > ### Comment · Reviewer_gxpW · 2022-11-21
> > > **Response to authors**
> > >
> > > Many thanks to the authors for their thoughtful response. I find the answers to my concerns quite persuasive. Especially the point about the non-availability of the ProofWriter dataset generation code, and the added flexibility of being able to adjust parameters etc by providing a new dataset. Accordingly I am happy to raise my score to 6.

---

### Official Review · Reviewer_GFj1 · 2022-10-25

**Confidence:** 3
**Correctness:** 3
**Technical Novelty And Significance:** 2
**Empirical Novelty And Significance:** 2
**Recommendation:** 5

**Clarity, Quality, Novelty And Reproducibility:**

The work is well written and clear, however, there exists another line of work which released a first order logic dataset [1]. So, it is unclear as to the originality of this paper. A comparison with new literature is needed.

Analysis of proofs is a novel aspect of this work. However, it is not clear what extent of intermediate steps is needed to generate workable proofs. Fig 2 says that there is a one-to-one correspondence between the conclusion of each proof step and the sentences in the chain-of-thought. This implies that the user is giving proofs or significant hints to generate the proof.




**Strength And Weaknesses:**

Strengths
● The paper is well structured and easy to read.
● Releasing a first order logic dataset for the community to test LLMs reasoning abilities.

Weakness
● Although GPT-3 is well known in the community, InstructGPT is something not popularly known and it would have helped to have a brief description as to what tasks InstructGPT can perform, along with its pre-training data.
● No mention of the number of entities modeled in the ontology.
● In the results, it would have been nice to know how many samples in the test-bed are used, and how many are valid/atomic/misleading.
● How is this work different from [1]
● A similar work to test reasoning of LLMs is done in [2], where it was established that LLMs cannot reason, which is in contradiction with the proposal in this paper.

[1]Han, Simeng, et al. "Folio: Natural language reasoning with first-order logic."
arXiv preprint arXiv:2209.00840 (2022).

[2]Valmeekam, Karthik, et al. "Large Language Models Still Can't Plan (A Benchmark for
LLMs on Planning and Reasoning about Change)." arXiv preprint arXiv:2206.10498 (2022).

**Summary Of The Paper:**

The paper releases a first order logic dataset, PRONTOQA, to evaluate the reasoning capabilities of LLMs using chain-of-thought prompting. The intermediate steps of reasoning, which are called chain-of-thought, can help in formal analysis of proofs. They also analyze results of GPT3 and  INSTRUCTGPT on this dataset.

**Summary Of The Review:**

This is an interesting line of work on improving sequential reasoning. It creates a new dataset to facilitate reasoning and analysis of proofs. Although promising, it is not clear if the burden put on the user for generating prompts at every reasoning step is practical. Further, the generated proof is a validation of the system or the user input.

The discussion of the reasoning quality with respect to ontology quality is interesting. However, if ontology is available, why not use it directly as a combination of neuro-symbolic reasoning? This aspect is not explored in depth.

The evaluations are well described.

---

> ### Author Response · Authors · 2022-11-15
> **Response to Reviewer GFj1**
>
> Thank you for the feedback and helpful comments! We would greatly appreciate if the reviewer would let us know if some of the revisions were insufficient and we can further iterate on the paper during the revision period.
>
> 1. “How is this work different from [Han et al. 2022]? A similar work to test reasoning of LLMs is done in [Valmeekam et al. 2022], where it was established that LLMs cannot reason, which is in contradiction with the proposal in this paper.”
>
> We thank the reviewer for bringing the missing related work to our attention. We have added them to the Related Work section. The evaluation in Valmeekam et al. 2022 is a planning task that requires abilities in addition to pure reasoning, which is the focus of our work. We added the following sentence to the Related Work section to address this: “Valmeekam et al. (2022) found that LLMs had difficulty with a fairly simple planning task, but it is not clear whether this was due to an inability to reason or other abilities instrumental in planning, such as world modeling, keeping track of state changes, and reasoning about events that occur sequentially in time. Our work aims to address this gap.”
>
> One important point that we wish to emphasize is that the main contribution of our work is not the dataset, but the analysis that is facilitated by the dataset, where we are able to formally inspect and study the predicted chains-of-thought of language models. Our analysis generalizes to other simple datasets. The analysis code is open source and freely available to anyone who wishes to apply it to their own data. To stress this point for readers, we add the following sentence to the second paragraph in the Introduction: “We emphasize here that while the dataset is an important contribution of this paper, the main contribution is the analysis that is facilitated by the dataset.”
>
> FOLIO, while similar to our proposed dataset in some respects, has a number of properties that make it less appropriate for our analysis:
> - FOLIO is human-annotated and so we are not able to modify the data generation process to remedy the above issues. Our exploration benefited from the flexibility and controllability that is afforded by working with our own dataset, such as the ability to add distractors to mitigate heuristics.
> - Each proof in FOLIO contains a mix of several different rules of deduction, as well as a mix of unary and binary predicates (e.g. “x is a cat” and “x is warm-blooded” are unary predicates whereas “x likes y” and “x has y” are binary). We endeavored to perform a more focused study where we would be able to measure differences in the model’s ability to reason over implication vs other logical connectives, or over binary vs unary predicates.
> - One of the main efforts in our analysis is to measure the confounding effect of knowledge acquired from pretraining during reasoning, and we explicitly added a variable in our data generation to produce examples that were either consistent with the real world (true), inconsistent (false), or neither (fictional). However, FOLIO only contains examples consistent with the real world. But this could be addressed by replacing nouns with fictional words.
>
> 2. “It is not clear if the burden put on the user for generating prompts at every reasoning step is practical. Further, the generated proof is a validation of the system or the user input… If ontology is available, why not use it directly as a combination of neuro-symbolic reasoning? This aspect is not explored in depth.”
>
> The purpose of the ontology is to facilitate the generation of the proofs which, in turn, are used to generate each example in the dataset. The ontology is not available to the model at test-time. The ontology, proofs, and resulting question-answering examples are generated automatically, without the need for user input.
>
> 3. InstructGPT is not as well known as GPT-3 in the community.
>
> We added the following sentence to the footnote on page 2: “InstructGPT is the model resulting from fine-tuning GPT-3 via reinforcement learning from human feedback.”
>
> 4. “No mention of the number of entities modeled in the ontology.”
>
> We added the following sentence to the paragraph titled “Ontology generation” in Section 3: “Since ontologies are randomly generated, they vary in size from as few as 3 concepts to as many as 10.”

---

> > ### Author Response · Authors · 2022-11-15
> > **Response to Reviewer GFj1 (continued)**
> >
> > 5. “In the results, it would have been nice to know how many samples in the test-bed are used, and how many are valid/atomic/misleading.”
> >
> > The number of samples is provided in section 5.1. To make this a bit clearer, we rephrased the sentence to: “For each combination of variables, we run the model on 400 examples generated from the testbed…”
> >
> > To address the second question, we added the following sentences to section 5.3: “[In] the 5-hop experiments with fictional ontology, 93.2% of proof steps are strictly-valid, 2.4% are broadly-valid, and 5.9% are invalid… [In] the 5-hop experiments with fictional ontology, 2.4% of proof steps are non-atomic, even though all steps in the few-shot examples are atomic.”

---

> > ### Comment · Reviewer_GFj1 · 2022-11-18
> > **Acknowledging the authors response, connections to planning / sequential reasoning tasks**
> >
> > The response of the authors is noted.
> >
> > Poor results in Valmeekam et al. (2022) are mostly due to reasoning  as other factors are controlled in PDDL style planning -- "inability to reason or other abilities instrumental in planning, such as world modeling, keeping track of state changes, and reasoning about events that occur sequentially in time". Nevertheless, appreciate the response.

---

> > > ### Author Response · Authors · 2022-11-19
> > > **Response to Reviewer GFj1 (reply)**
> > >
> > > We thank the reviewer for their feedback and timely response.
> > >
> > > In the arXiv version of Valmeekam et al. (2022) posted Oct 29th 2022, in section 5.3, they describe a prompt that would test "reasoning about plan execution," which would control for several variables such as "plan generation," "robustness to plan formulation," etc, and more closely test the reasoning behavior that our work aims to test. However, in section 6, results for that particular prompt are not actually provided, whereas results for "plan generation" and "robustness to plan formulation" are provided, among others. But even the "reasoning about plan execution" as described in their paper still contains confounding abilities, in addition to "pure reasoning" (which we aim to test). These confounding variables include "world modeling, keeping track of state changes, and reasoning about events that occur sequentially in time," and are not controlled for by "plan generation," "robustness to plan formulation," or the other planning-related abilities tested by Valmeekam et al.
> > >
> > > Furthermore, their experiments utilize a Blocksworld environment, which may not suitably control for the confounding factor of knowledge acquired from pretraining. For example, we found that the model's reasoning ability was very good when given question-answering examples that are consistent with their pretraining (i.e. true ontology), but this was not the case in the results of Valmeekam et al.
> > >
> > > To address this concern, we modified the relevant sentences in the Related Work section:
> > > "[Valmeekam et al] found that LLMs had difficulty with a fairly simple planning task, but it is not clear whether this was due to an inability to reason or other abilities instrumental in planning, such as world modeling, keeping track of state changes, and reasoning about events that occur sequentially in time. This is despite their controlling for other variables involved in planning, such as plan generation, robustness to goal formulation, among others. They experimented with examples in a ``Blocksworld'' environment, a significant portion of which the LLM can acquire from pretraining. Our work aims to address this gap."

---

### Official Review · Reviewer_LZJG · 2022-10-25

**Confidence:** 4
**Correctness:** 3
**Technical Novelty And Significance:** 1
**Empirical Novelty And Significance:** 3
**Recommendation:** 6

**Clarity, Quality, Novelty And Reproducibility:**

The paper is generally easy to follow, though there are a few issues with clarity:

- It's not super clear what the difference between strictly-valid non-atomic and broadly-valid is, a more detailed explanation may be helpful.

- In Figure 3, how is the accuracy on the y-axis measured? Is it the percentage of deduction steps that are correct, or the percentage of questions where all deduction steps are correct?

- The bars in Figure 4 are quite confusing and it's unclear how to interpret the plot.

The work is low in novelty, but it illuminates some interesting observations about the reasoning abilities of LLMs and introduces a dataset that I think will be useful to the community.

**Strength And Weaknesses:**

### Strength:

- The paper introduces a dataset for synthetic reasoning that can be useful for probing the reasoning behavior of LLMs. The authors take care to remove spurious heuristic solutions from the task setup, and restrict the complexity of the dataset while still maintaining a few knobs of question difficulty that one can use.
- The paper explores several interesting notions of validity, atomicity, and misleading steps, which seems to be a fairly comprehensive perspective for studying reasoning behavior in models.
- The paper makes interesting observations regarding the tendency of the model to skip steps and output misleading steps which are irrelevant to the final answer, and its ability to perform basic deductions well.

### Weakness:

- The paper makes a number of observations that have been similarly made in other papers, such as 1) the model relies on pretrained knowledge to perform reasoning (also discussed in [1] and [2]), 2) longer proofs are more challenging and correct answers can sometimes contain misleading steps (which are well known and have been noted in many papers evaluating LLMs and chain-of-thought on reasoning tasks), and 3) sentence ordering affects performance (also discussed in [3]). This limits the contributions of this work.

- Several of the directions mentioned in future work seems well within the scope of the current paper and should probably be included to strengthen the current paper. For example, seeing if including prompt examples containing misleading steps would improve performance is an immediate follow-up to the observation that most errors come from misleading steps. The beam search of deduction steps is also a reasonable analysis to include in the current work in my opinion.

>Our work shows that relying on CoT prompting is not sufficient for more complex reasoning, such as in mathematical domains.

I don't see how this work shows the CoT prompting is insufficient for reasoning in mathematical domains.

[1] https://arxiv.org/abs/2202.12837

[2] https://arxiv.org/abs/2202.07206

[3] https://arxiv.org/abs/2104.08786

**Summary Of The Paper:**

This paper introduces a PrOntoQA dataset and uses it to study the performance of chain-of-thought prompting of large language models. PrOntoQA is a synthetic reasoning dataset where each example is generated from an ontology along with the proof steps. The questions are answerable from repeated applications of the modus ponens deduction rule, which makes the reasoning process fairly simple, with the main driver of complexity coming from the number of deduction steps needed. The dataset is motivated by the desire to easily evaluate the quality of the intermediate chain-of-thought outputs, and to disentangle the reasoning performance that comes from pretraining world knowledge from the pure logical reasoning ability. The authors find that GPT-3 performs very well on individual deduction steps but can fail to output the correct chain of steps needed to solve the problem when there are multiple deduction paths possible. The authors further find that using fictional or false words instead of the true entities significantly hinders performance, thus illustrating the model's reliance on its pretraining knowledge and understanding of the entities to correctly perform the reasoning.

**Summary Of The Review:**

This paper introduces a dataset that I think will be useful to the community, and the analysis illuminates some interesting observations about the reasoning abilities of LLMs. However the observations are not very novel or surprising, and the manuscript could improve in its clarity in certain parts.

---

> ### Author Response · Authors · 2022-11-15
> **Response to Reviewer LZJG**
>
> Thank you for the feedback and helpful comments! We would greatly appreciate if the reviewer would let us know if some of the revisions were insufficient and we can further iterate on the paper during the revision period.
>
> 1. “The paper makes a number of observations that have been similarly made in other papers: the model relies on pretrained knowledge to perform reasoning (also discussed in [1] and [2])”
>
>   The work in the papers referenced by the reviewer focus broadly on in-context learning and they do not look at reasoning specifically, and so it is unclear whether their results would generalize to reasoning or to what extent. For example, [2] focuses their attention on the model’s ability to compute arithmetic operations, which is a highly specific kind of reasoning restricted to mathematical domains, and it is very possible that the model has learned to better generalize using rules of deduction such as modus ponens. We found that the model is still able to reason even in the fictional ontology, where it isn’t relying on pretrained knowledge. While [1] finds that randomly replacing labels barely affects model performance, we note a significant drop in performance when moving from the true ontology to the fictional ontology.
>   To better place our work in the context of these papers, we add the following sentences to the Related Work section:
>   “Recent work has also examined in-context learning and found that performance on certain tasks is highly dependent on the prompt [2, 3]. However, they focused on sentiment classification and simple arithmetic tasks, and it is not clear if their results generalize to reasoning. The LLM could feasibly use retrieval, rather than reasoning, to perform those tasks. Our experiments on the fictional ontology show that the model is able to reason even when there is nothing to retrieve from.”
>
> [1] Sewon Min, Xinxi Lyu, Ari Holtzman, Mikel Artetxe, Mike Lewis, Hannaneh Hajishirzi, Luke Zettlemoyer. Rethinking the Role of Demonstrations: What Makes In-Context Learning Work? arXiv, 2022.
> [2] Yasaman Razeghi, Robert L. Logan IV, Matt Gardner, and Sameer Singh. Impact of pretraining term frequencies on few-shot reasoning. arXiv, 2022.
> [3] Yao Lu, Max Bartolo, Alastair Moore, Sebastian Riedel, and Pontus Stenetorp. Fantastically ordered prompts and where to find them: Overcoming few-shot prompt order sensitivity. arXiv, 2021.
>
> 2. “The paper makes a number of observations that have been similarly made in other papers: longer proofs are more challenging and correct answers can sometimes contain misleading steps”
>
>   Our experiments demonstrate not only that correct answers sometimes contain misleading steps, but that misleading steps are a main source of error in the model’s reasoning.
>
> 3. “The paper makes a number of observations that have been similarly made in other papers: sentence ordering affects performance (also discussed in [3])”
>
>   We believe the reviewer misinterpreted the phrase “sentence ordering” to refer to the order of the in-context examples. Rather, we were referring to the ordering of the sentences *within* each in-context example, and so [3] is not very relevant to our work. To avoid this confusion, we substituted the term “sentence ordering” with “traversal direction” throughout the paper, as it is less overloaded.
>
> [3] Yao Lu, Max Bartolo, Alastair Moore, Sebastian Riedel, and Pontus Stenetorp. Fantastically ordered prompts and where to find them: Overcoming few-shot prompt order sensitivity. arXiv, 2021.
>
> 4. “Several of the directions mentioned in future work seems well within the scope of the current paper and should probably be included to strengthen the current paper.”
>
>   We added section A.7 to the appendix exploring whether our results generalize to self-consistency prompting [4] and depth-first search. We found that neither self-consistency nor DFS improves proof accuracy on this task. We also find that for examples that the model gets wrong, the model actually assigns higher overall probability to the incorrect proof than to the correct proof. This suggests that prompting/decoding strategies that encourage the model to produce more globally-optimal outputs (e.g. beam search) would not remedy the model’s shortcomings in proof planning.
>
> [4] Xuezhi Wang, Jason Wei, Dale Schuurmans, Quoc V. Le, Ed H. Chi, and Denny Zhou. Self-
> consistency improves chain of thought reasoning in language models. arXiv, 2022.

---

> > ### Author Response · Authors · 2022-11-15
> > **Response to Reviewer LZJG (continued)**
> >
> > 5. “I don't see how this work shows the CoT prompting is insufficient for reasoning in mathematical domains.”
> >
> >   Mathematical reasoning is a strict superset of the reasoning tested in this paper, relying on many more deduction rules beyond modus ponens, and there are proof steps in mathematical proofs with much higher branching factor. Thus, proof planning is much more instrumental in the correctness of mathematical proofs. To clarify this argument in the paper, we revised the sentence in the second paragraph of the conclusion:
> >   “Our work shows that CoT prompting is not sufficient for more complex reasoning, such as in mathematical domains, since the reasoning tested in this work is a strict subset of that of general mathematical reasoning. Mathematical proofs contain steps with much higher branching factor, where robust proof planning is instrumental.”
> >
> > 6. “It's not super clear what the difference between strictly-valid non-atomic and broadly-valid is, a more detailed explanation may be helpful.”
> >
> >   We add the following sentence to clarify this point to the second paragraph of Section 4: “For example, given the premises, ‘Cats are carnivores’ and ‘Carnivores are mammals,’ the step with conclusion ‘Cats are mammals’ is broadly-valid since an additional deduction rule is required to prove it: given \forall x(f(x) \to g(x)) and \forall x(g(x) \to h(x)), conclude \forall x(f(x) \to h(x)). Notice that this is distinct from a strictly-valid non-atomic step since this conclusion is not provable via repeated applications of modus ponens.”
> >
> > 7. The y-axis in figure 3 is unclear.
> >
> >   We added the following sentence to the caption of figure 3: “We emphasize that ‘proof accuracy’ indicates the fraction of proofs (not proof steps) that are considered correct according to our metrics.”
> >
> > 8. The dark bars in figure 4 are hard to interpret.
> >
> >   We revised the caption of figure 4 to clarify: “Each bar is subdivided into six darker bars according to the types of proof steps that appear in the predicted chains-of-thought. For example, the dark red bar corresponding to ‘invalid steps’ indicates the proportion of incorrect proofs that contain an invalid step. The dark blue bar corresponding to ‘invalid steps’ indicates the proportion of correct proofs that contain an invalid step.”

---

> > > ### Comment · Reviewer_LZJG · 2022-11-22
> > > **Rebuttal acknowledgement and response**
> > >
> > > Thank you for the detailed rebuttal and clarifications!
> > >
> > > I'm not very convinced that some of the contributions are very new in light of prior work. E.g. both [1] and [2] discuss the usage of pretrained information in solving reasoning tasks, and they don't claim that *only* pretrained information is used. I agree that the synthetic dataset offers a clearer look at the problem in a different setting than prior works and that this can be valuable, though the conclusions are still in line with existing insights and are not too surprising.
> > >
> > > Thanks for clearing up the confusion around sentence ordering. I wonder if this result says something general about reasoning or is it specific to this exact dataset?
> > >
> > > Thanks for adding the new experiments about beam search, I think those are interesting. However, my main suggestion (seeing if including prompt examples containing misleading steps would improve performance is an immediate follow-up to the observation that most errors come from misleading steps) does not seem to be tried. My concern here was that I want to make sure that the main limitation that this paper raises (i.e. unable to recover from misleading steps, which contributes to poor proof planning) would not be easily solved by having a more informative prompt. The suggested experiment seems not too difficult to implement, but I am open to a discussion from the author on this point or some other experiment to address this question. I would be open to recommending acceptance if this part is sufficiently addressed.

---

> > > > ### Author Response · Authors · 2022-11-22
> > > > **Response to Reviewer LZJG (reply)**
> > > >
> > > > We again thank you for your detailed and helpful feedback.
> > > >
> > > > We actually did run those experiments where the prompt contains examples with misleading steps in section A.7.2 (which we referred to as "depth-first search" since each CoT is now a trace of a DFS on the proof space). Thus, for each in-context example, there is a 50% chance it will contain a misleading step. We found that the proof accuracy is not significantly different in this setting (0.55 vs 0.545). Given the opportunity, we will further clarify this in the final version of the paper.

---

> > > > > ### Comment · Reviewer_LZJG · 2022-12-08
> > > > > **Thanks for the response**
> > > > >
> > > > > I will update my score to a 6

---

### Official Review · Reviewer_jd47 · 2022-10-27

**Confidence:** 3
**Correctness:** 4
**Technical Novelty And Significance:** 3
**Empirical Novelty And Significance:** 3
**Recommendation:** 6

**Clarity, Quality, Novelty And Reproducibility:**

Clarity: Excellent.  Most of the definitions in the paper are given with lots of relevant examples, which are very helpful for understanding. The graphs and explanations in the method description and experimental performance are also detailed. So the clarity is excellent.

Quality: Good.  The purpose of the research in this paper is clear (i.e., to propose a form of dataset for evaluating LLM). The proposed approach regarding the generation of thought chains is very interesting and the final experiment is well done. So the quality is good.

Novelty: Excellent.  There are almost no valid assessment datasets to evaluate LLM reasoning ability, and the proposal of this paper has a positive effect on the refinement of LLM reasoning ability later. So the proposal of this paper is novelty.

Reproducibility: Avarage.  The proposed dataset generation logic in this paper is clear and step-by-step. From ontology tree to generating proof and context, and finally generating CoT, quary and label from the proof. where the example generation syntax of PRONTOQA is also relatively simple. However, the large model GPT tested in the experiment is not available to most people. So the reproducibility is average.


**Strength And Weaknesses:**

Strengths:
1) The generative process of example in PROTOQA is logical, systematic, interpretable and traceable. The process of logical reasoning is very clear.
2) The interesting experiment is sufficient enough and shows us the reasoning condition of the big model.
3) The concepts in the paper are explained clearly, and examples make it easier for readers to understand
4) This paper allows the neural system and the symbolic system to cooperate in a better way

Weaknesses:
The context in the dataset may be monotonous to some extent. Will the future work experiment on more diversified natural languages?


**Summary Of The Paper:**

The paper uses the information generated by the symbol system to help the neural system model to better reason and test CoT reasoning ability, which creates a Dataset PRONTOQA which aim at testing the reasoning ability of big model, and proposes a new method to measure the correctness of reasoning steps rather than the accuracy of reasoning answers. The experiment reveal some interesting findings of CoT.


**Summary Of The Review:**

The process of example generation in PROTOQA is logical, systematic, interpretable and traceable. The process of logical reasoning is also very clear. This paper uses the information generated by this symbolic system to help neural system models reason better and test CoT reasoning ability. It is very innovative. Very much in line with the ICRL conference requirements. Acceptance of this paper is recommended.

---

> ### Author Response · Authors · 2022-11-15
> **Response to Reviewer jd47**
>
> Thank you for the feedback and helpful comments! We would greatly appreciate if the reviewer would let us know if some of the revisions were insufficient and we can further iterate on the paper during the revision period.
>
> 1. The language in the dataset looks too artificial and/or unrealistic.
>
>   The simplicity of the sentences in the dataset is intentional in order to ensure that the predicted output from the LLM is easily semantically-parseable. This facilitates the formal analysis of the chain-of-thought. There are many instances of simple or synthetic datasets being very useful and illuminating in NLP research, such as [1], [2], [3] (this list is very non-comprehensive). And we hope that our dataset will similarly serve to help illuminate the reasoning abilities of LLMs.
>
> [1] Jason Weston, Antoine Bordes, Sumit Chopra, and Tomás Mikolov. Towards ai-complete question answering: A set of prerequisite toy tasks. ICLR 2016.
> [2] Avia Efrat, Or Honovich, and Omer Levy. Lmentry: A language model benchmark of elementary language tasks. arXiv, 2022.
> [3] Yuhuai Wu, Felix Li, and Percy Liang. Insights into Pre-training via Simpler Synthetic Tasks. arXiv, 2022.
>
> 2. “The large model GPT tested in the experiment is not available to most people. So the reproducibility is average.”
>
>   To address this, we uploaded all model outputs to the repository containing our code. We have attached an anonymized copy of this repository as supplementary material. This repository also contains the code to generate examples from the dataset and to perform the analysis and produce all figures in the paper with a single Python command. We added the following Reproducibility Statement: “All our experiments in the main text were run using the OpenAI API on September 9th, 10th, and 11th, 2022. The self-consistency experiment was run on October 29th and 30th, and the DFS experiment was run on November 16th (see section A.7). For the sake of reproducibility of the analysis, all model outputs, the code for data generation, and the analysis code are freely available with a permissive open-source license at [anonymized]. The command python analyze_results.py produces all figures used in this paper.”
>   We edited the footnote on page 1: “All analysis code, data, data generation scripts, and model outputs are available at [anonymized].”
>   In addition to our experiments on InstructGPT, we also evaluated the original GPT-3 model (davinci) which is stable.

---

### Author Response · Authors · 2022-11-15
**Common Response to Reviewers**

We thank all reviewers for the helpful feedback and comments! We will address common concerns below and also respond to each reviewer individually.

We added a number of sections to the appendix in the course of our revisions. If given an additional page, we would be able to move the more important sections to the main text.

We have also changed the title of the paper to "Language Models are Greedy Reasoners: A Systematic Formal Analysis of Chain-of-Thought."

1. A number of authors raised concerns about the novelty of our work with respect to prior work.

  One important point that we wish to emphasize is that the main contribution of our work is not the dataset, but the analysis that is facilitated by the dataset, where we are able to formally inspect and study the predicted chains-of-thought of language models. Our analysis generalizes to other simple datasets, including ProofWriter. The analysis code is open source and freely available to anyone who wishes to apply it to their own data. To stress this point for readers, we add the following sentence to the second paragraph in the Introduction:
  “We emphasize here that while the dataset is an important contribution of this paper, the main contribution is the analysis that is facilitated by the dataset.”

  We also acknowledge the similarity of the proposed dataset PrOntoQA with ProofWriter, and we agree that a deeper comparison is warranted in the paper. We added a paragraph and a table at the beginning of the Related Work section to provide this comparison, but to expand further here: There are a number of reasons why we chose to create a new dataset rather than perform our analysis on ProofWriter.
  - While the ProofWriter data is freely available, their code to generate the data is not, and our exploration benefited from the flexibility and controllability that is afforded by working with our own dataset, such as the ability to add distractors to mitigate heuristics.
  - Each proof in ProofWriter contains a mix of implication and conjunction, as well as a mix of unary and binary predicates (e.g. “x is a cat” and “x is warm-blooded” are unary predicates whereas “x likes y” and “x has y” are binary). We endeavored to perform a more focused study where we would be able to measure differences in the model’s ability to reason over implication vs conjunction, or over binary vs unary predicates.
  - One of the main efforts in our analysis is to measure the confounding effect of knowledge acquired from pretraining during reasoning, and we explicitly added a variable in our data generation to produce examples that were either consistent with the real world (true), inconsistent (false), or neither (fictional). However, ProofWriter does not make this distinction, with examples containing all three types of rules.
  These concerns apply to FOLIO as well, but in addition, since FOLIO is human-annotated, we are not able to modify the data generation process to remedy the above issues.

---

> ### Author Response · Authors · 2022-11-15
> **Common Response to Reviewers (continued)**
>
> 2. Some reviewers brought to our attention missing related work.
>
>   We thank these reviewers for doing so! We have added the following sentences to the Related Work section:
>   “Similar to our approach, [1] converts logical forms into fairly simple natural language using templates. However, The examples in these datasets are largely consistent with the real-world, and so they may confound measuring reasoning ability with retrieval ability.
>   …
>   [5] found that LLMs had difficulty with a fairly simple planning task, but it is not clear whether this was due to an inability to reason or other abilities instrumental in planning, such as world modeling, keeping track of state changes, and reasoning about events that occur sequentially in time. Our work aims to address this gap.
>   …
>   Recent work has also examined in-context learning and found that performance on certain tasks is highly dependent on the prompt [3, 4]. However, they focused on sentiment classification and simple arithmetic tasks, and it is not clear if their results generalize to reasoning.”
>
>   [1] Gregor Betz. Critical thinking for language models. arXiv, 2020.
>   [2] Antonia Creswell and Murray Shanahan. Faithful reasoning using large language models. arXiv, 2022.
>   [3] Yao Lu, Max Bartolo, Alastair Moore, Sebastian Riedel, and Pontus Stenetorp. Fantastically ordered prompts and where to find them: Overcoming few-shot prompt order sensitivity. arXiv, 2021.
>   [4] Yasaman Razeghi, Robert L. Logan IV, Matt Gardner, and Sameer Singh. Impact of pretraining term frequencies on few-shot reasoning. arXiv, 2022.
>   [5] Karthik Valmeekam, Alberto Olmo Hernandez, Sarath Sreedharan, and Subbarao Kambhampati. Large language models still can’t plan (A benchmark for LLMs on planning and reasoning about change). arXiv, 2022.
>   [6] Xuezhi Wang, Jason Wei, Dale Schuurmans, Quoc V. Le, Ed H. Chi, and Denny Zhou. Self-
> consistency improves chain of thought reasoning in language models. arXiv, 2022.
>
> 3. A number of reviewers also raised the concern that our results are not reproducible since it relies on querying the OpenAI API.
>
>   To address this, we uploaded all model outputs to the repository containing our code. We have attached an anonymized copy of this repository as supplementary material. This repository also contains the code to generate examples from the dataset and to perform the analysis and produce all figures in the paper with a single Python command. We added the following Reproducibility Statement: All our experiments in the main text were run using the OpenAI API on September 9th, 10th, and 11th, 2022. The self-consistency experiment was run on October 29th and 30th, and the DFS experiment was run on November 16th (see section A.7). For the sake of reproducibility of the analysis, all model outputs, the code for data generation, and the analysis code are freely available with a permissive open-source license at [anonymized]. The command python analyze_results.py produces all figures used in this paper.”
>   We edited the footnote on page 1: “All analysis code, data, data generation scripts, and model outputs are available at [anonymized].”
>   One concern is that the InstructGPT models (e.g. text-davinci-002) are not stable, and that OpenAI may change them in the future. In addition to our experiments on InstructGPT, we also evaluated the original GPT-3 model (davinci) which is stable.

---

### Decision · Program_Chairs · 2023-01-20

**Decision:**

Accept: poster

**Justification For Why Not Higher Score:**

The insights are not new or surprising.

**Justification For Why Not Lower Score:**

the systematic study is valuable and well-done.

**Metareview: Summary, Strengths And Weaknesses:**

The paper aims to investigate Chain of thought reasoning by proposing a new synthetic dataset called PrOntoQA, where each example is generated from a synthetic world model represented in first-order logic. and using that parse the generated chain-of-thought into symbolic proofs for formal analysis.

Strength:
- The generative process of example in PROTOQA is logical, systematic, interpretable and traceable. The process of logical reasoning is very clear.
- The experiments are extensive on the reasoning condition of the big models.
- The concepts in the paper are explained clearly, and examples make it easier for readers to understand
- This paper allows the neural system and the symbolic system to cooperate in a better way

Weakness:
- The context in the dataset may be monotonous to some extent.
- The paper makes a number of observations that have been similarly made in other papers



**Note From Pc:**

if the above contains the word "oral" or "spotlight" please see: "oral" presentation means -> notable-top-5% and "spotlight" means -> notable-top-25%. As stated in our emails, we are disassociating presentation type from AC recommendations

**Summary Of Ac-Reviewer Meeting:**

We discussed the strength and weaknesses of the paper and what would be the reasons to accept or reject it. One of the reviewers said that although they gave it a 6 they really meant an accept and the 6 was their calibration. The other reviewer that has given 5 had concerns about the work that are really the case for the whole approach to LLMs and they not being able to public to investigate the weights. Although the paper shares insights that are not new, the systematic investigation is valuable. Therefore, we decided to accept the paper.